# A Unified Framework for Quantized and Continuous Strong Lottery Tickets

## Abstract

The Strong Lottery Ticket Hypothesis (SLTH) asserts that sufficiently overparameterized, randomly initialized neural networks contain sparse subnetworks that, even without any training, can match the performance of a small trained network on a given dataset. A key mathematical tool in the theoretical study of SLTH has been the Random Subset Sum Problem (RSSP). The SLTH has recently been extended to the quantized setting, where the network weights are sampled from a discrete set rather than from a continuous interval. These new results are however far from those in arbitrary-precision setting in several ways. In this work, we provide an analysis of the RSSP in the discrete setting, and use it to derive tight SLTH guarantees in the quantized case. Our analysis obtain tight bounds on the failure probability of finding a strong lottery ticket in the quantized regime, providing an exponential improvement over previous results. Most importantly, it unifies the literature by showing that both approximate representations in the continuous setting and exact representations in quantized settings naturally emerge as limiting cases of our results. This perspective not only sharpens existing bounds but also provides a cohesive framework that simultaneously handles approximation and rounding errors.

## 1 Introduction

Deep neural networks (DNNs) have achieved remarkable success across a wide range of machine learning tasks. However, their rapidly increasing size and complexity introduce significant challenges for both training and deployment. This has motivated extensive research on neural network pruning as a means to reduce model size and computational cost while preserving performance. One prominent theoretical framework in this direction is the Strong Lottery Ticket Hypothesis (SLTH), which posits that sufficiently overparameterized, randomly initialized neural networks contain sparse subnetworks—so-called strong lottery tickets—that can match the performance of a smaller trained network on a given task, without any training. We refer the reader to Section 2 for details on the SLTH literature. The emergence of SLTH has motivated numerous results demonstrating that a sufficiently overparameterized network can be pruned to simulate a given smaller "target" network[1] (Zhou et al., 2019; Orseau et al., 2020; Malach et al., 2020; Pensia et al., 2020; Burkholz, 2022a; Diffenderfer & Kailkhura, 2021; Sreenivasan et al., 2022; Kumar & Natale, 2025). A central combinatorial problem underlying the SLTH theory is the Random Subset Sum Problem (RSSP) (Lueker, 1998), which has been used to quantify the overparameterization required for a large random network to contain such performant subnetworks.

Most prior analyses in SLTH assume arbitrary precision for network weights, this is largely because the classical theory of RSSP has been developed in a continuous setting. Nevertheless, several works have addressed the finite-precision regime. In particular, Diffenderfer & Kailkhura (2021); Sreenivasan et al. (2022) consider versions of the problem where networks are restricted to finite-precision weights. Diffenderfer & Kailkhura (2021) demonstrates—both **theoretically and empirically**—that sufficiently overparameterized networks with binary weights can approximate any network of a given size. Building on this, Sreenivasan et al. (2022) further strengthened the theoretical results (see Section 2). However, these works remain limited in scope, since the target networks

---

[1]The target network is merely a proof artifact and is not known in practice; in fact, identifying such a network is NP-hard.

still use continuous weights while the approximating networks are constrained to binary ones. Subsequently, by leveraging elegant results by Borgs et al. (2001) on the Number Partitioning Problem (NPP), Kumar & Natale (2025) has managed to overcome such restriction, obtaining a partial extension of the RSSP theory to the quantized regime, where both the initial and target networks have weights drawn from discrete sets. Specifically, Kumar & Natale (2025) leveraged a formal connection between the RSSP and the Number Partitioning Problem (NPP), obtaining bounds for the exact representation of a quantized target network. A key limitation of such analysis is that it yields only an inverse-polynomial decay of the failure probability. This should be contrasted with the inverse-exponential decay of previous SLTH results in the continuous setting (Pensia et al., 2020; Orseau et al., 2020). Moreover, previous SLTH literature quantified the success probability in terms of the admissible approximation error $\epsilon$ of the subnetwork, whereas Kumar & Natale (2025) only provided an estimate of the probability of finding an exact subnetwork, leaving open the question of how these guarantees extend to approximate representations. Our goal is to provide stronger theoretical guarantees that support the existing empirical observations in this direction, and to unify the continuous and quantized lines of research.

| Paper | Quantized | Approx. | Exact Representation | Exponential Probability |
|---|---|---|---|---|
| Malach et al. (2020) | ✗ | ✓ | ✗ | ✗ |
| Pensia et al. (2020) | ✗ | ✓ | ✗ | ✓ |
| Orseau et al. (2020) | ✗ | ✓ | ✗ | ✓ |
| Diffenderfer & Kailkhura (2021) | ✓ | ✓ | ✗ | ✗ |
| Kumar & Natale (2025) | ✓ | ✗ | ✓ | ✗ |
| Ours | ✓ | ✓ | ✓ | ✓ |

Table 1: A qualitative comparison to prior work: Our results simultaneously cover approximate representations in the continuous setting and exact representations in the quantized setting, with previous results arising as special cases of our framework. In addition, we establish an exponentially decaying failure probability in the quantized regime. There are certain caveats to this taxonomy; a detailed justification for the classification is provided in Appendix C.

**Our contributions.** We close the aforementioned gaps by proving new, sharp bounds for the discrete RSSP that are tailored to the quantized SLTH setting. Specifically, we generalize the results of Cunha et al. (2023) on the continuous RSSP to the discrete setting, allowing us to establish quantized SLTH results with exponentially small failure probability in the number of precision bits—an exponential improvement over the earlier inverse-polynomial bound. Furthermore, our analysis unifies previous results by simultaneously handling both rounding and approximation errors, making a fundamental step towards broadening the practical relevance of the SLTH theory. More importantly, both approximate representations in the continuous setting, as in Malach et al. (2020); Pensia et al. (2020); Orseau et al. (2020), and exact representations in a quantized setting, as in Diffenderfer & Kailkhura (2021); Sreenivasan et al. (2022); Kumar & Natale (2025), arise as special cases in the limit of our results. Our results can be summarized by the following simplified, informal theorem (refer to the formal statements for full generality).

**Theorem** (Informal version of Theorem 3). *With high probability, a depth-$2\ell$ randomly-initialized network $N_{in}$ of width $\mathcal{O}\big(d\log(1/\delta)\big)$ whose weights are represented using $\log 1/\delta$ bits of precision can be pruned to $\epsilon$ approximate any target network $N_t$ with $\ell$ layers of width at most $d$.*

Table 1 presents a comparison of our results with previous works. These results advance the theoretical understanding of quantized strong lottery tickets in several ways:

1. they extend the scope of SLTH theory to include both quantized and $\epsilon$-approximate representations,

2. they strengthen previous probabilistic guarantees from inverse-polynomial to exponential decay, and

3. both approximate representations in the continuous setting, and exact representations in a quantized setting, can be obtained as special cases in the limit of our results.

**Paper organization.** In Section 2, we review prior work on SLTH and the role of the RSSP in its theoretical foundations. In Section 3, we extend the analysis of Cunha et al. (2023) to the quantized RSSP and prove our main probabilistic bounds. In Section 4, we use these results to establish new SLTH guarantees in the quantized setting, including both exact and $\epsilon$-approximate cases. We conclude in Section 5 with a discussion of implications, open problems, and future directions.

## 2 RELATED WORK

The Lottery Ticket Hypothesis (LTH) was first proposed by Frankle & Carbin (2019), who showed that every dense neural network contains a sparse subnetwork that can be trained from scratch to achieve the same accuracy as the original dense model. Soon after, a series of surprising empirical results (Zhou et al., 2019; Ramanujan et al., 2020; Wang et al., 2019) demonstrated that one can find subnetworks within large randomly initialized neural networks that already perform well on a given task, without any weight updates. This line of work led to the Strong Lottery Ticket Hypothesis (SLTH), which posits that sufficiently overparameterized random networks contain sparse subnetworks that approximate small trained networks without training.

Theoretical progress on the SLTH began with Malach et al. (2020), who proved that any feed-forward network of depth $\ell$ and width $d$ can be approximated by pruning a random network of depth $2\ell$ and width $\mathcal{O}(d^5\ell^2)$. Subsequent works (Orseau et al., 2020; Pensia et al., 2020) improved this to $\mathcal{O}(d \log(d\ell))$, while Burkholz (2022a) established a different construction showing that a depth-$(\ell+1)$ network suffices to approximate a depth-$\ell$ target, with certain trade-offs in the width. Further extensions broadened the scope of the SLTH to convolutional architectures (Burkholz, 2022b) and equivariant networks (Ferbach et al., 2022).

A complementary direction of research investigates quantization, the process of reducing weight precision. Empirical studies (Han et al., 2015) demonstrated that trained networks can often be quantized significantly without loss of accuracy, and Diffenderfer & Kailkhura (2021) provided both empirical and theoretical evidence for a quantized SLTH, introducing binary subnetworks that approximate real-valued networks. They showed that a network of width $d$ and depth $\ell$ can be approximated within error $\epsilon$ by a binary network of width $\mathcal{O}(\ell d^{3/2}/\epsilon + \ell d \log(\ell d/\epsilon))$ and depth $2\ell$. Later, Sreenivasan et al. (2022) exponentially improved these bounds, showing that binary networks of depth $\Theta(\ell \log(d\ell/\epsilon))$ and width $\Theta(d \log^2(d\ell/\epsilon))$ suffice. Recent work has also connected quantized subnetworks to universal approximation guarantees (Hwang et al., 2024), while practical advances such as mixed-precision quantization (Carilli, 2020; Younes Belkada, 2022) explore more hardware-efficient designs.

Most recently, Kumar & Natale (2025) established the first sharp theoretical guarantees for the quantized SLTH by leveraging connections between the RSSP and the Random Number Partitioning Problem (RNPP). They showed that pruning can yield exact quantized subnetworks, and characterized the optimal trade-off between overparameterization and weight precision. However, their failure probability bounds decay only inverse-polynomially, and their analysis was limited to exact representations—gaps that motivate the present work.

**Subset Sum Problem (SSP).** The SSP is a classical NP-complete problem in computational complexity (Garey & Johnson, 1979), where the task is to decide whether a given target value $z$ can be expressed as the sum of a subset of a given set of numbers. Its random variant, the Random Subset Sum Problem (RSSP), has been extensively investigated in the context of combinatorial optimization (Lueker, 1982; 1998; Borst et al., 2023). More recently, Cunha et al. (2023) provided a simpler and more elementary proof of these classical results, while Kumar & Natale (2025) extended the analysis to the discrete setting, building on the seminal work of Borgs et al. (2001) on the Number Partitioning Problem. The relevance of the RSSP to the theory of the Strong Lottery Ticket Hypothesis has since been highlighted in several works (Pensia et al., 2020; Burkholz, 2022a; Kumar & Natale, 2025).

## 3 DISCRETE RANDOM SUBSET SUM

The **Subset Sum Problem** is a fundamental problem in computational complexity theory. It is a well-known **NP-complete** problem Garey & Johnson (1979). Its randomized variant, the **Random**

**Subset Sum Problem (RSSP)**, has recently attracted attention in the machine learning community, due to its application in the theory of **SLTH** (Pensia et al., 2020; Burkholz, 2022a; Kumar & Natale, 2025). A key result in the theory of the RSSP was established by Lueker (1998), stated as Theorem 1.

**Theorem 1** (Lueker (1998)). *Consider the set of random variables $X_1, X_2, \ldots, X_n$ sampled uniformly from $U[-1, 1]$. If $n > C \log\left(\frac{1}{\epsilon}\right)$, then with probability at least $1 - \epsilon$, for any $z \in \left[-\frac{1}{2}, \frac{1}{2}\right]$, there exists $S \subseteq [n]$ such that*

$$\left| \sum_{i \in S} X_i - z \right| \leq \epsilon.$$

This result has played a central role in all the SLTH results discussed in Section 2. An elementary proof of Theorem 1 was provided by Cunha et al. (2023). In this section, we present a discrete variant of their proof, which will be particularly important for our analysis of quantization in the context of the SLTH in Section 4. We start by defining RSSP in the discrete setting.

**Definition 1.** *Consider the set of random variables $X_1, X_2, \ldots, X_n$ sampled uniformly from $\{-M, -M+1, \ldots, M-1, M\}$ for some positive integer $M$ and target $z \in \{-M, -M+1, \ldots, M-1, M\}$. Let $\Delta \in \mathbb{N} \cup \{0\}$ be given. The RSSP is the problem of finding a subset $S \subseteq [n]$ such that*

$$\left| \sum_{i \in S} X_i - z \right| \leq \Delta.$$

Let $f_t : \mathbb{Z} \to \{0, 1\}$ be the indicator for the event "$z$ is $\Delta$ approximated" at time $t \in \mathbb{N}$, i.e.,

$$f_t(z) = \mathbb{I}_{\exists S \subseteq [t] \,:\, |\sum_{i \in S} X_i - z| \leq \Delta},$$

with $f_t(0) = 1$ by definition. One of the fundamental properties of $f_t$, first observed by Lueker (1998), is that it satisfies the following recurrence relation

$$f_{t+1}(z) = f_t(z) + (1 - f_t(z)) f_t(z - X_{t+1}). \tag{1}$$

Define the volume $v_t$ as

$$v_t = \sum_{i=-M}^{M} f_t(i).$$

The objective of our analysis is to understand the growth of $v_t$ as a function of $t$. In particular, we are interested in the time at which $v_t$ reaches $2M + 1 - \Delta$, since this marks the point where the entire set $\{-M, \ldots, M\}$ can be $\Delta$-approximated. We denote by $\tau$ the first time that the process $v_t$ reaches at least $2M + 1 - \Delta$.

**Theorem 2.** *Let $\Delta \in \{0, \ldots, M\}$. There exist constants $C > 0$ and $\kappa > 0$ such that for every $t \geq C \log \frac{M+1}{\Delta+1}$, it holds that*

$$\mathbb{P}(\tau \leq t) \geq 1 - 2 \exp\left[ -\frac{1}{\kappa t} \left( t - C \log \frac{M+1}{\Delta+1} \right)^2 \right].$$

Theorem 2 asserts that if $t \geq C \log \frac{M+1}{\Delta+1}$, then the entire set can be $\Delta$-approximated with high probability. To establish this result, we discretize the analysis in Cunha et al. (2023). Below, we provide a sketch of the proof; full details are deferred to Appendix A.

*Proof Sketch.* The recurrence relation (Equation 1) encapsulates the structure of the RSSP and serves as a powerful tool. Using this relation, we first establish that

$$\mathbb{E}(v_{t+1} | X_1, X_2, \ldots, X_t) \geq v_t + \frac{v_t}{4} \left( 1 - \frac{v_t}{2M} \right). \tag{2}$$

This is explicitly shown in Lemma 2, Appendix A. Equation 2 plays a central role in the proof, as the argument essentially reduces to analyzing the growth of the stochastic process $v_t$. Note that, for the RSSP to be solvable for any target, it is essential to estimate the number of samples required for $v_t$ to approach $2M + 1$, thereby ensuring that the entire set can be well approximated. The analysis

in Appendix A shows that $v_t$ grows quickly to $M + 1$ and then slowly rises to $2M + 1$. We analyze these two phases of growth of $v_t$ separately. For the first phase, define $\tau_1$ as

$$\tau_1 = \min\{t \geq 0 : v_t > M + 1\}.$$

Lemma 3, Appendix A characterizes how $v_t$ grows in the first phase by proving that

$$\mathbb{P}(v_{t+1} \geq v_t(1 + \beta) \mid X_1, \ldots, X_t, t \leq \tau_1) \geq 1 - \frac{7}{8(1 - \beta)},$$

for any $\beta \in (0, 1/8)$. Using this, we estimate the time required for $v_t$ to grow to $M + 1$ with high probability. In particular Lemma 4, Appendix A shows that

$$\mathbb{P}(\tau_1 \leq t) \geq 1 - \exp\left(-\frac{2p_\beta^2}{t}\left(t - \frac{i^*}{p_\beta}\right)^2\right) \qquad \text{if} \qquad t \geq \frac{1}{1 - \frac{1}{8(1+\beta)}}\left\lceil\frac{\log\frac{M+1}{2\Delta+1}}{\log(1 + \beta)}\right\rceil.$$

We then similarly analyze the growth of $v_t$ in the second phase, i.e., where the volume grows from $M + 1$ to $2M - \Delta + 1$. Consider the process $w_t = (2M + 1) - v_{\tau_1+t}$. Lemma 5, Appendix A shows that

$$\mathbb{E}(w_{t+1}|X_1, \ldots, X_{t+\tau_1}) \leq w_t\left(1 - \frac{1}{4}\left(1 - \frac{w_t}{2M + 1}\right)\right).$$

Let $\tau_2$ the first time that $w_t$ gets smaller than or equal to $2M - \Delta + 1$, that is

$$\tau_2 = \min\{t \geq 0 : w_t \leq 2M - \Delta + 1\}.$$

Lemma 6, Appendix A shows that

$$\mathbb{P}(\tau_2 \leq t) \geq 1 - \frac{1}{2M + 1 - \Delta}\left(\frac{7}{8}\right)^t.$$

We then tie the two parts to estimate $\mathbb{P}(\tau \leq t)$. In particular we show that

$$\mathbb{P}(\tau \leq t) = \mathbb{P}(\tau_1 + \tau_2 \leq t)$$

$$\geq 1 - \exp\left(-\frac{1}{15^2 t}\left(t - 30\left\lceil\frac{\log\frac{M+1}{2\Delta+1}}{\log\frac{17}{16}}\right\rceil\right)^2\right) - \frac{1}{2M + 1 - \Delta}\left(\frac{7}{8}\right)^{t/2}$$

$$\geq 1 - 2\exp\left[-\frac{1}{\kappa t}\left(t - C'\log\frac{M + 1}{\Delta + 1}\right)^2\right],$$

which proves the statement of Theorem 2. See Appendix A for details. $\qquad\square$

Having established a robust probabilistic guarantee for the discrete Random Subset Sum Problem, we now shift our focus to its application in the context of the Strong Lottery Ticket Hypothesis. The preceding analysis provides the core mathematical tool required to rigorously analyze the existence of strong lottery tickets within a quantized framework. In this next section, we will build upon the foundation of Theorem 2 to demonstrate how an overparameterized neural network with discrete weights can be effectively pruned to approximate a target network, thereby connecting the abstract combinatorial results to tangible implications for quantized neural networks.

## 4 QUANTIZATION AND SLTH

In this section, we use the results from Section 3 to prove quantized SLTH results. Our strategy is to follow Pensia et al. (2020); Kumar & Natale (2025), while leveraging our new Theorem 2 for quantizing the use of RSSP. We begin by defining some notation. Scalars are denoted by lowercase letters such as $w$, $y$, etc. Vectors are represented by bold lowercase letters, e.g., $\mathbf{v}$, and the $i^{\text{th}}$ component of a vector $\mathbf{v}$ is denoted by $v_i$. Matrices are denoted by bold uppercase letters such as $\mathbf{M}$. If a matrix $\mathbf{W}$ has dimensions $d_1 \times d_2$, we write $\mathbf{W} \in \mathbb{R}^{d_1 \times d_2}$. For a vector $\mathbf{v}$, we use $\|\mathbf{v}\|$ to denote it's $\ell_2$ norm. We define the finite set $S_\delta := \{-1, -1+\delta, -1+2\delta, \ldots, 1\}$, where $\delta = 2^{-k}$ for

some $k \in \mathbb{N}$. A real number $b$ is said to have precision $\delta$ if $b \in S_\delta$. We denote the $d$-fold Cartesian product of $S_\delta$ by $S_\delta^d$; that is,

$$S_\delta^d := \underbrace{S_\delta \times \cdots \times S_\delta}_{d \text{ times}}.$$

We use $C, C'$ etc., to denote positive absolute constants. A notation table (Table B) is provided in Appendix B for the reader's convenience.

**Definition 2.** *For any integers $d_0, ..., d_\ell > 0$ let $\mathcal{F}_{d_0,...,d_\ell}$ be the class of $\ell$-layer neural networks $f : \mathbb{R}^{d_0} \to \mathbb{R}^{d_\ell}$ defined as*

$$f(\mathbf{x}) := \mathbf{W}_\ell \sigma(\mathbf{W}_{\ell-1} \cdots \sigma(\mathbf{W}_1 \mathbf{x})), \tag{3}$$

*where $\mathbf{W}_i \in S_\delta^{d_i \times d_{i-1}}$ for $i = 1, \ldots, \ell$, $\mathbf{x} \in \mathbb{R}^{d_0}$, and $\sigma : \mathbb{R} \to \mathbb{R}$ is a nonlinear activation function. For a vector $\mathbf{x}$, the expression $\mathbf{v} = \sigma(\mathbf{x})$ denotes component-wise application: $v_i = \sigma(x_i)$.*

For a network as defined in Equation 3, we shall denote $d = \max\{d_1, \ldots d_\ell\}$. The entries of the matrices $\mathbf{W}_i$ are referred to as the weights or parameters of the network. In this work, we assume all activation functions are ReLU, i.e., $\sigma(x) = \max(0, x)$. This assumption is made for simplicity; the results can be extended to a broader class of activation functions as discussed in Burkholz (2022a).

We will say a neural network is $\delta$-quantized if each of its weight is sampled from $S_\delta$. Our objective is to approximate a *target* $\delta$-quantized $\ell$-layer neural network $f$ by suitably pruning an overparameterized $2\ell$-layer $\delta$-quantized network $g$. For a neural network

$$g(\mathbf{x}) = \mathbf{M}_{2\ell} \sigma(\mathbf{M}_{2\ell-1} \cdots \sigma(\mathbf{M}_1 \mathbf{x})).$$

The pruned network $\hat{g}$ is defined as:

$$\hat{g}(\mathbf{x}) = (\mathbf{S}_{2\ell} \odot \mathbf{M}_{2\ell}) \sigma((\mathbf{S}_{2\ell-1} \odot \mathbf{M}_{2\ell-1}) \cdots \sigma((\mathbf{S}_1 \odot \mathbf{M}_1) \mathbf{x})),$$

where each $\mathbf{S}_i$ is a binary pruning mask with the same dimensions as $\mathbf{M}_i$, and $\odot$ denotes element-wise multiplication. Hence, the goal is to find masks $\mathbf{S}_1, \mathbf{S}_2, \ldots, \mathbf{S}_{2\ell}$ such that $f$ can be $\epsilon$ approximated by $\hat{g}$. As will be shown, the problem reduces to solving a collection of RSSPs, which, throughout this section, we assume are to be solved with tolerance $\Delta$ (See Section 3). We now state our main result.

**Theorem 3.** *Let $d_0, ..., d_\ell$ be any integers greater than 0, $\epsilon$ be any real number in $(0, 1)$ and $\delta = \epsilon/2\Delta N_T$ where $N_T = \sum_{i=1}^{\ell} d_{i-1} d_i$ and $\Delta \in \mathbb{Z}^+$. Consider a randomly initialized $2\ell$-layered neural network*

$$g(\mathbf{x}) = \mathbf{M}_{2\ell} \sigma(\mathbf{M}_{2\ell-1} \ldots \sigma(\mathbf{M}_1 \mathbf{x})),$$

*where every weight is drawn uniformly from $S_\delta$. Let $n > \log \frac{\frac{1}{\delta}+1}{\Delta+1}$. Assume $\mathbf{M}_{2i}$ has dimension*

$$d_i \times C d_{i-1} n,$$

*and $\mathbf{M}_{2i-1}$ has dimension*

$$C d_{i-1} n \times d_{i-1}.$$

*Then, with probability at least*

$$1 - 4d^2 \ell \exp(-C'n),$$

*for every $f \in \mathcal{F}_{d_0,...,d_\ell}$ it holds*

$$\min_{\mathbf{S}_i \in \{0,1\}^{d_i \times d_{i-1}}} \sup_{\|x\| \leq 1} \|f(\mathbf{x}) - (\mathbf{S}_{2\ell} \odot \mathbf{M}_{2\ell}) \sigma((\mathbf{S}_{2\ell} \odot \mathbf{M}_{2\ell}) \ldots \sigma((\mathbf{S}_1 \odot \mathbf{M}_1) \mathbf{x}))\| \leq \varepsilon.$$

For convenience, we define $\log \frac{\frac{1}{\delta}+1}{\Delta+1}$ as the *overparameterization factor*.

*Remark* 1. The assumptions made in our analysis can be further relaxed. In particular, the requirement of uniformly distributed weights can be removed via a standard rejection sampling argument Lueker (1998). Moreover, the restriction to ReLU activations can also be relaxed, as shown in Burkholz (2022a).

We follow the strategy introduced in Pensia et al. (2020) to prove Theorem 3. We first show the result for a single weight.

**Lemma 1** (Approximating a weight). *Let $\epsilon$ be any real number in $(0,1)$ and $\delta = \epsilon/2\Delta$, where $\Delta \in \mathbb{Z}^+$. Let $g : \mathbb{R} \to \mathbb{R}$ be a randomly initialized network of the form $g(x) = \mathbf{v}^T \sigma(\mathbf{u}x)$, where $\mathbf{v}, \mathbf{u} \in S_\delta^{2n}$ with $n \geq C \log \frac{\frac{1}{\delta}+1}{\Delta+1}$, and all $v_i, u_i$'s are drawn i.i.d. uniformly from $S_\delta$. Then with probability at least*

$$1 - 4\exp\left(-\frac{1}{\kappa n}\left(n - C' \log \frac{\frac{1}{\delta}+1}{\Delta+1}\right)^2\right),$$

*we have for any $w \in S_\delta$*

$$\exists\, \mathbf{s}, \mathbf{s}' \in \{0,1\}^{2n} : \sup_{x:|x|\leq 1} |wx - (\mathbf{v} \odot \mathbf{s})^T \sigma((\mathbf{u} \odot \mathbf{s}')(x))| \leq \epsilon.$$

*Proof.* We decompose $wx$ as $wx = \sigma(wx) - \sigma(-wx)$ and w.l.o.g.[2] assume $w \geq 0$ (the reasoning for $w < 0$ is analogous). Let

$$\mathbf{v} = \begin{bmatrix} \mathbf{b} \\ \mathbf{d} \end{bmatrix}, \quad \mathbf{u} = \begin{bmatrix} \mathbf{a} \\ \mathbf{c} \end{bmatrix}, \quad \mathbf{s} = \begin{bmatrix} \mathbf{s_1} \\ \mathbf{s_2} \end{bmatrix}, \quad \mathbf{s}' = \begin{bmatrix} \mathbf{s_1}' \\ \mathbf{s_2}' \end{bmatrix},$$

where $\mathbf{a}, \mathbf{b}, \mathbf{c}, \mathbf{d} \in S_\delta^n$, $\mathbf{s_1}, \mathbf{s_2} \in \{0,1\}^n$. One can thus verify that

$$(\mathbf{v} \odot \mathbf{s})^T \sigma((\mathbf{u} \odot \mathbf{s}')x) = (\mathbf{b} \odot \mathbf{s_1})^T \sigma(\mathbf{a} \odot \mathbf{s_1}'x) + (\mathbf{d} \odot \mathbf{s_2})^T \sigma(\mathbf{c} \odot \mathbf{s_2}'x).$$

**Step 1: Pre-processing $\mathbf{a}$.** Let

$$\mathbf{a}^+ = \max\{0, \mathbf{a}\}$$

be the vector obtained by pruning all negative entries of $\mathbf{a}$. Then $\mathbf{a}^+$ contains $n$ i.i.d. random variables uniformly distributed over non negative entries of $S_\delta$. Since we assumed $w \geq 0$, for $x \leq 0$ we have $\sigma(wx) = 0$ and $\mathbf{b}^T \sigma(\mathbf{a}^+x) = 0$. We thus focus on $x > 0$:

$$\sigma(wx) = wx, \quad \mathbf{b}^T \sigma(\mathbf{a}^+x) = \sum_i b_i a_i^+ x.$$

We simply choose $\mathbf{s_1}'$ such that $\mathbf{a}^+ = \mathbf{a} \odot \mathbf{s_1}'$.

**Step 2: Pruning $\mathbf{a}$ via SUBSETSUM.** Consider the random variables $Z_i = b_i a_i^+$. Note that solving the Subset Sum Problem in an integer setting where numbers are sampled from $\{-M, \ldots, M\}$ and solving it when numbers are sampled from $\{-1, \ldots \delta, 2\delta, \ldots, 1\}$ is equivalent: the only difference is a scaling factor. Note that the numbers $\{Z_i\}_i$ are of precision $\delta^2$, and they are not uniformly distributed (See Kumar & Natale (2025)). However, for $n$ large enough, a constant fraction of these samples is uniformly distributed (Theorem 5, Appendix F). Hence by Theorem 2, as long as

$$n \geq C \log \frac{\frac{1}{\delta}+1}{\Delta+1},$$

with probability at least

$$1 - 2\exp\left(-\frac{1}{\kappa n}\left(n - C \log \frac{\frac{1}{\delta}+1}{\Delta+1}\right)^2\right),$$

we can choose a subset of $\{Z_i\}$ such that

$$\forall w \in S_\delta, \, \left|w - \mathbf{b}^T(\mathbf{s_1} \odot \mathbf{a}^+)\right| < \Delta\delta.$$

Hence, it holds

$$\forall w \in S_\delta, \, \sup_{x \in [-1,1]} \left|\sigma(wx) - \mathbf{b}^T \sigma((\mathbf{s_1} \odot \mathbf{a}^+)x)\right| < \Delta\delta.$$

**Step 3: Pre-processing $\mathbf{c}$.** Let

$$\mathbf{c}^- = \min\{0, \mathbf{c}\}$$

---

[2]Without loss of generality.

be the vector obtained by pruning all positive entries of $\mathbf{c}$. Then $\mathbf{c}^-$ contains $n$ i.i.d. random variables uniformly distributed over non positive entries of $S_\delta$. We simply choose $\mathbf{s_2}'$ such that $\mathbf{c}^- = \mathbf{c} \odot \mathbf{s_2}'$.

**Step 4: Pruning c via SUBSETSUM.** For $x \geq 0$, $\sigma(-wx) = 0$ and $\mathbf{d}^T \sigma(\mathbf{c}^- x) = 0$. Moreover, pruning $\mathbf{c}^-$ further does not affect the equality. Thus, we only consider the case $x < 0$.

For $x < 0$, one has $-\sigma(-wx) = wx$ and $\sigma(\mathbf{c}^- x) = \mathbf{c}^- x$, so

$$\mathbf{d}^T \sigma(\mathbf{c}^- x) = (\mathbf{d}^T \mathbf{c}^-) x.$$

Applying Theorem 2 again, with $n \geq C \log \frac{\frac{1}{\delta}+1}{\Delta+1}$, with probability at least

$$1 - 2 \exp\left(-\frac{1}{\kappa n}\left(n - C \log \frac{\frac{1}{\delta}+1}{\Delta+1}\right)^2\right),$$

there exists $\mathbf{s}_2 \in \{0,1\}^n$ such that

$$\forall w \in S_\delta, \quad \sup_{x \in [-1,1]} \left|-\sigma(-wx) - \mathbf{d}^T \sigma((\mathbf{s}_2 \odot \mathbf{c}^-)x)\right| < \Delta\delta.$$

**Step 5: Tying it all together.** Recall that we assumed, w.l.o.g., $w \geq 0$. By the above reasoning and a union bound, both events hold with probability at least

$$1 - 4 \exp\left(-\frac{1}{\kappa n}\left(n - C' \log \frac{\frac{1}{\delta}+1}{\Delta+1}\right)^2\right).$$

We thus get that

$$\inf_{\mathbf{s} \in \{0,1\}^{2n}} \sup_{|x| \leq 1} \left|wx - (\mathbf{v} \odot \mathbf{s})^T \sigma((\mathbf{u} \odot \mathbf{s}')x)\right|$$

$$= \inf_{\mathbf{s}_1, \mathbf{s}_2} \sup_{|x| \leq 1} \left|wx - (\mathbf{b} \odot \mathbf{s_1})^T \sigma(\mathbf{a}^+ x) - (\mathbf{d} \odot \mathbf{s_2})^T \sigma(\mathbf{c}^- x)\right|$$

$$= \inf_{\mathbf{s}_1, \mathbf{s}_2} \sup_{|x| \leq 1} \left|\sigma(wx) - \sigma(-wx) - (\mathbf{b} \odot \mathbf{s_1})^T \sigma(\mathbf{a}^+ x) - (\mathbf{d} \odot \mathbf{s_2})^T \sigma(\mathbf{c}^- x)\right|$$

$$\leq \inf_{\mathbf{s}_1} \sup_{|x| \leq 1} \left|\sigma(wx) - (\mathbf{b} \odot \mathbf{s_1})^T \sigma(\mathbf{a}^+ x)\right| + \inf_{\mathbf{s}_2} \sup_{|x| \leq 1} \left|-\sigma(-wx) - (\mathbf{d} \odot \mathbf{s_2})^T \sigma(\mathbf{c}^- x)\right|$$

$$\leq 2\Delta\delta.$$

Since $\delta = \varepsilon/2\Delta$, the result follows. $\square$

Having proved Lemma 1, we now give a sketch of the proof of Theorem 3. For the detailed proof, see Appendix D.

*Sketch of proof of Theorem 3.* The idea is to follow the argument of Pensia et al. (2020), which extends the approximation from a single weight to the entire network. The extension from a single weight to an entire network proceeds in successive stages. Lemma 1 shows that a single quantized weight can be simulated with high probability using a sufficiently wide two-layer construction, with approximation error at most $\epsilon$. First, this guarantee is then lifted to the level of a neuron in Lemma 7, Appendix D: since a neuron applies several weights to its inputs, each weight is handled separately via Lemma 1, and a union bound ensures that all incoming weights are simultaneously approximated with exponentially high probability, so that the entire neuron behaves as desired up to error $\epsilon$. Next, Lemma 8, Appendix D extends this argument to a full layer. The weight sharing trick introduced by Pensia et al. (2020) is used (See Appendix D for details) and another union bound guarantees that the layer as a whole is approximated uniformly over the input space, with only an additive accumulation of error. Finally, the approximation is propagated across all $\ell$ layers of the network. Starting from the first layer, one iteratively applies the layer-level result to approximate each subsequent layer, carefully tracking the errors through the nonlinearities. The error propagation analysis shows that the total deviation remains within the prescribed tolerance $\epsilon$, while the failure probability across all layers is still exponentially small due to the bounds established at the weight, neuron, and layer levels. This yields Theorem 3, which asserts that with high probability, a randomly initialized $2\ell$-layer quantized network can be pruned to $\epsilon$-approximate any target $\ell$-layer quantized network. Further details are provided in Appendix D. $\square$

## 4.1 PREVIOUS RESULTS AS SPECIAL CASES

A striking feature of our results is that both approximate representations in the continuous setting (Malach et al., 2020; Pensia et al., 2020; Orseau et al., 2020), and exact representations in a quantized setting (Diffenderfer & Kailkhura, 2021; Kumar & Natale, 2025), can be obtained as special cases in the limit of our results. To see this, consider the over overparameterization factor

$$\log \frac{\frac{1}{\delta} + 1}{\Delta + 1}. \tag{4}$$

Notice that an error $\epsilon$ in the continuous setting corresponds to an error $\Delta \cdot \delta$ in the quantized setting. Substituting this into Equation 4, we obtain

$$\log \frac{\frac{1}{\delta} + 1}{\Delta + 1} = \log \frac{\frac{1}{\delta} + 1}{\frac{\epsilon}{\delta} + 1}$$
$$= \log \frac{1 + \delta}{\epsilon + \delta}.$$

In the limit $\delta \to 0$, which corresponds to the continuous setting, the overparameterization factor reduces to $\log \frac{1}{\epsilon}$, matching the expression in Pensia et al. (2020); Orseau et al. (2020). Conversely, setting $\Delta = 0$, corresponding to exact representation in the quantized setting, yields $\log \left( \frac{1}{\delta} + 1 \right)$, which aligns with the overparameterization in Kumar & Natale (2025).

## 4.2 QUANTIZED NEURON ACTIVATIONS

Our results apply to standard neural networks whose weights are sampled from a discrete set. However, the activations in these networks are assumed to have arbitrary precision, which does not fully reflect the behavior of real-world neural networks, since computers always operate with finite precision. In contrast, Kumar & Natale (2025) established their results under mixed-precision assumptions, where neuron activations are also discretized. Interestingly, our results remain valid under these assumptions as well. In particular, consider a mixed-precision network as in Kumar & Natale (2025), where activations in even-numbered layers have precision $\delta$ and those in odd-numbered layers have precision $\delta^2$. Under this setup, Lemmas 1, 7, and 8 remain valid. Using Lemma 8, the error propagation can then be done while accounting for this mixed-precision assumption, leading to a version of Theorem 3 adapted to the mixed-precision setting.

## 5 CONCLUSION

In this work, we established sharp bounds for the discrete Random Subset Sum Problem (RSSP) and applied them to the quantized Strong Lottery Ticket Hypothesis (SLTH). Our main contribution is to show that both approximate representations and exact representations in quantized settings naturally arise as limiting cases of our analysis. This unifying perspective brings together previously separate lines of work under a single framework. In addition, our results demonstrate that the failure probability of finding a sparse subnetwork in a randomly initialized, quantized neural network decays exponentially with overparameterization—a substantial improvement over the inverse-polynomial guarantees in prior work. By simultaneously addressing approximation and rounding errors, our framework broadens the scope of SLTH theory to more realistic finite-precision regimes, while recovering and strengthening earlier results. Overall, this work highlights the deep interplay between combinatorial problems such as RSSP and foundational questions in deep learning, and marks an important step toward a cohesive theory of strong lottery tickets across both continuous and quantized settings.

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

## A  RSSP RESULTS

Let's start by recalling the setup from Section 3. Consider the set of random variables $X_1, X_2, \ldots, X_n$ sampled uniformly from $\{-M, -M+1, \ldots, M-1, M\}$ for some positive integer $M$ and target $z \in \{-M, -M+1, \ldots, M-1, M\}$. Let $\Delta \in \mathbb{N} \cup \{0\}$. The Random Subset Sum Problem (RSSP) is the problem of finding a subset $S \subseteq [n]$ such that

$$\big|\sum_{i \in S} X_i - z\big| \leq \Delta.$$

Let $f_t : \mathbb{Z} \to \{0, 1\}$ be the indicator for the event $z$ is $\Delta$ approximated at time $t \in \mathbb{N}$, i.e.,

$$f_t(z) = \mathbb{I}_{\exists S \subseteq [t] \,:\, |\sum_{i \in S} X_i - z| \leq \Delta}.$$

It is clear that $f_t$ follows the following recurrence relation

$$f_{t+1}(z) = f_t(z) + (1 - f_t(z))f_t(z - X_{t+1}).$$

Define $v_t$ as

$$v_t = \sum_{i=-M}^{M} f_t(i),$$

with $f_t(0) = 1$ by definition.

**Lemma 2.** *For all $0 \leq t \leq n$, it holds that*

$$\mathbb{E}(v_{t+1}|X_1, X_2, \ldots, X_t) \geq v_t + \frac{v_t}{4}\left(1 - \frac{v_t}{2M}\right).$$

*Proof.* Consider

$$\mathbb{E}(v_{t+1}|X_1, X_2, \ldots, X_t) = \mathbb{E}\left(\sum_{i=-M}^{M} f_{t+1}(i)|X_1, X_2, \ldots, X_t\right)$$

$$= \mathbb{E}\left(\sum_{i=-M}^{M} (f_t(i) + (1 - f_t(i))f_t(i - X_{t+1}))|X_1, X_2, \ldots, X_t\right)$$

$$= \frac{1}{2M+1} \sum_{j=-M}^{M} \sum_{i=-M}^{M} f_t(i) + (1 - f_t(i))f_t(i - j)$$

$$= \sum_{i=-M}^{M} f_t(i) + \frac{1}{2M+1} \sum_{j=-M}^{M} \sum_{i=-M}^{M} (1 - f_t(i))f_t(i - j)$$

$$= v_t + \frac{1}{2M+1} \sum_{i=-M}^{M} \left((1 - f_t(i)) \sum_{j=-M}^{M} f_t(i - j)\right)$$

$$= v_t + \frac{1}{2M+1} \sum_{i=-M}^{M} (1 - f_t(i)) \sum_{k=i-M}^{i+M} f_t(k)$$

where $k = i - j$. Hence we have

$$\mathbb{E}(v_{t+1}|X_1, X_2, \ldots, X_t) \geq v_t + \frac{1}{2M+1} \sum_{i=-\frac{M}{2}+\mu}^{i=\frac{M}{2}+\mu} \left((1 - f_t(i)) \sum_{k=-\frac{M}{2}+\mu}^{k=\frac{M}{2}+\mu} f_t(k)\right)$$

for some $\mu \in \{-M/2, \ldots, M/2\}$ (by range restriction). Now $\exists \, \mu^* \in \{-M/2, \ldots, M/2\}$ such that

$$\sum_{k=-\frac{M}{2}+\mu}^{k=-\frac{M}{2}+\mu} f_t(k) \geq \frac{v_t}{2}.$$

Choose $\mu = \mu^*$. Hence we get

$$\mathbb{E}(v_{t+1}|X_1, X_2, \ldots, X_t) \geq v_t + \frac{1}{2M}\frac{v_t}{2} \sum_{i=-\frac{M}{2}+\mu^*}^{i=\frac{M}{2}+\mu^*} (1 - f_t(i))$$

$$\geq v_t + \frac{v_t}{4}\left(1 - \frac{v_t}{2M}\right).$$

$\square$

The main results depend on the analysis of how $v_t$ grows with $t$. It turns out that $v_t$ grows quickly to $M + 1$ and then slowly rises to $2M + 1$. Define $\tau_1$ as

$$\tau_1 = \min\{t \geq 0 : v_t > M + 1\}.$$

**Lemma 3.** *Given $\beta \in (0, 1/8)$, let $p_\beta = 1 - \frac{7}{8(1-\beta)}$. For all integers $0 \leq t < \tau_1$ it holds that*

$$\mathbb{P}(v_{t+1} \geq v_t(1 + \beta) \mid X_1, \ldots, X_t, t \leq \tau_1) \geq p_\beta.$$

*Proof.* Define the process

$$\tilde{v} = \sum_{i=-M}^{M} \left( f_t(i) + (1 - f_t(i)) f_t(i - X_{t+1}) \mathbb{I}_{\{-M,\dots,M\}} \right)$$

Clearly $\tilde{v} \leq v_{t+1}$ and $\tilde{v} \leq 2v_t$. The bound in Lemma 2 also holds for $\tilde{v}$. Hence we have

$$\mathbb{P}(v_{t+1} \geq v_t(1 + \beta) \mid X_1, \dots, X_t, t \leq \tau_1) \geq \mathbb{P}(\tilde{v} \geq v_t(1 + \beta) \mid X_1, \dots, X_t, t \leq \tau_1).$$

Now recall the reverse Markov's inequality: for any random variable $X$ such that $0 \leq X \leq b$ and $0 \leq a \leq \mathbb{E}[X]$, we have

$$\mathbb{P}(X \geq a) \geq \frac{\mathbb{E}[X] - a}{b - a}.$$

Using reverse Markov's inequality, we have

$$\mathbb{P}(\tilde{v} \geq v_t(1 + \beta) \mid X_1, \dots, X_t, t \leq \tau_1) \geq \frac{\mathbb{E}[v_t \mid X_1, \dots, X_t, t < \tau_1] - v_t(1 - \beta)}{2v_t - v_t(1 - \beta)}$$

$$\geq \frac{\frac{9}{8}v_t - v_t(1 + \beta)}{2v_t - v_t(1 + \beta)}$$

$$= \frac{\frac{9}{8} - (1 + \beta)}{1 - \beta}$$

$$= 1 - \frac{7}{8(1 - \beta)}.$$

$\square$

**Lemma 4.** *Let $t$ be an integer and given $\beta \in (0, 1/8)$, let $p_\beta = 1 - \frac{1}{8(1+\beta)}$ and $i^* = \left\lceil \frac{\log \frac{M+1}{2\Delta+1}}{\log(1+\beta)} \right\rceil$. If $t \geq i^*/p_\beta$, then*

$$\mathbb{P}(\tau_1 \leq t) \geq 1 - \exp\left( -\frac{2p_\beta^2}{t} \left( t - \frac{i^*}{p_\beta} \right)^2 \right).$$

*Proof.* Divide the domain $\{0, \dots, M\}$ into intervals

$$I_0 = \{1, \dots, 2\Delta + 1\},$$
$$I_i = \left\{ \lfloor (2\Delta + 1)(1 + \beta)^{i-1} \rfloor, \dots, \lfloor (2\Delta + 1)(1 + \beta)^i \rfloor \right\},$$
$$I_{i^*} = \left\{ \lfloor (2\Delta + 1)(1 + \beta)^{i^*-1} \rfloor, \dots, M + 1 \right\},$$

where $i^*$ is the smallest integer for which

$$\left\lfloor (2\Delta + 1)(1 + \beta)^{i^*} \right\rfloor \geq M + 1$$

$$\implies i^* = \left\lceil \frac{\log \frac{M+1}{2\Delta+1}}{\log(1 + \beta)} \right\rceil.$$

We are interested in the number of steps required for $v_t$ to exit the interval $I_i$ after entering it. By Theorem 3, this amount is majorised by a geometric random variable $Y_i \sim \text{Geom}(p_\beta)$. Therefore, we can conclude that $\tau_1$ is stochastically dominated by the sum of such variables, that is, for $t \in \mathbb{N}$, we have that

$$\mathbb{P}(\tau_1 \geq t) \leq \mathbb{P}\left( \sum_{i=1}^{i^*} Y_i \geq t \right). \tag{5}$$

Let $B_t \sim \text{Binomial}(t, p_\beta)$ be a binomial random variable. For the sum of geometric random variables, it holds that $\mathbb{P}\left( \sum_{i=1}^{i^*} Y_i \leq t \right) = \mathbb{P}(B_t \geq i^*)$. Since $\mathbb{E}(B_t) = tp_\beta$, the Hoeffding's bound

for binomial random variables implies that, for all $\lambda \geq 0$, we have that $\mathbb{P}(B_t \leq tp_\beta - \lambda) \leq \exp(-2\lambda^2/t)$. Setting $t$ such that $tp_\beta - \lambda = i^*$, we obtain that

$$\mathbb{P}\left(\sum_{i=1}^{i^*} Y_i \geq t\right) \leq \mathbb{P}(B_t \leq i^*)$$

$$\leq \exp\left(-\frac{2}{t}(tp_\beta - i^*)^2\right)$$

$$= \exp\left(-\frac{2p_\beta^2}{t}\left(t - \frac{i^*}{p_\beta}\right)^2\right)$$

which holds as long as

$$t \geq \frac{1}{p_\beta}\left\lceil\frac{\log\frac{M+1}{2\Delta+1}}{\log(1+\beta)}\right\rceil.$$

The result follows by applying this to Equation 5 and considering the complementary events. $\quad\square$

Now we study the second half of the process: from the moment the volume reaches $M+1$ up to the time it gets to $2M - \Delta + 1$. Consider the process $w_t = (2M+1) - v_{\tau_1+t}$.

**Lemma 5.** *For all $t \geq 0$, it holds that*

$$\mathbb{E}(w_{t+1}|X_1, \ldots, X_{t+\tau_1}) \leq w_t\left(1 - \frac{1}{4}\left(1 - \frac{w_t}{2M+1}\right)\right).$$

*Proof.* Consider

$$\mathbb{E}[w_{t+1}|X_1, \ldots, X_{t+\tau_1}] = \mathbb{E}[(2M+1) - v_{t+\tau_1+1}|X_1, \ldots, X_{t+\tau_1}]$$
$$= (2M+1) - \mathbb{E}(v_{t+\tau_1+1}|X_1, \ldots, X_{t+\tau_1})$$
$$\leq (2M+1) - v_{t+\tau_1}\left(1 + \frac{1}{4}\left(1 - \frac{v_{t+\tau_1}}{2M+1}\right)\right)$$
$$= (2M+1) - (2M+1-w_t)\left(1 + \frac{1}{4}\left(1 - \frac{2M+1-w_t}{2M+1}\right)\right)$$
$$= (2M+1) - \left[(2M+1-w_t) + (2M+1-w_t)\frac{1}{4}\frac{w_t}{2M+1}\right]$$
$$= w_t - (2M+1-w_t)\frac{1}{4}\frac{w_t}{2M+1}$$
$$= w_t - \frac{1}{4}w_t\left(1 - \frac{w_t}{2M+1}\right)$$
$$= w_t\left(1 - \frac{1}{4}\left(1 - \frac{w_t}{2M+1}\right)\right).$$

$\quad\square$

Let $\tau_2$ the first time that $w_t$ gets smaller than or equal to $2M - \Delta + 1$, that is

$$\tau_2 = \min\{t \geq 0 : w_t \leq 2M - \Delta + 1\}.$$

**Lemma 6.** *For all $t > 0$, it holds that*

$$\mathbb{P}(\tau_2 \leq t) \geq 1 - \frac{1}{2M+1-\Delta}\left(\frac{7}{8}\right)^t.$$

*Proof.* Using $2M + 1 - w_t = v_{t+\tau_1} > M + 1$ to Lemma 5, we get

$$\mathbb{E}(w_{t+1}|X_1, \ldots, X_{t+\tau_1}) \leq \frac{7}{8}w_t. \tag{6}$$

Moreover, from the conditional expectation theory, for any two random variables $X$ and $Y$, we have $\mathbb{E}[\mathbb{E}[X|Y]] = \mathbb{E}[X]$. Form this and Eq. 6 we get

$$\mathbb{E}[w_t] = \mathbb{E}[\mathbb{E}(w_{t+1}|X_1,\ldots,X_{t+\tau_1})] \leq \frac{7}{8}\mathbb{E}[w_{t-1}],$$

which, by recursion, yields that

$$\mathbb{E}[w_t] \leq \left(\frac{7}{8}\right)^t \mathbb{E}[w_0] \leq \frac{1}{2}\left(\frac{7}{8}\right)^t.$$

Finally, by Markov's inequality,

$$\mathbb{P}(\tau_2 \geq t) \leq \mathbb{P}(w_t \geq 2M + 1 - \Delta) \leq \frac{\mathbb{E}[w_t]}{2M+1-\Delta} \leq \frac{1}{2M+1-\Delta}\left(\frac{7}{8}\right)^t,$$

and the result follows from considering the complementary event. $\qquad\square$

Define $\tau = \tau_1 + \tau_2$, the first time at which the process $v_t$ reaches at least $2M + 1 - d$.

**Theorem 4.** *Let $\Delta \in \{0,\ldots,M\}$. There exist constants $C' > 0$ and $\kappa > 0$ such that for every $t \geq C'\log\frac{M+1}{\Delta+1}$, it holds that*

$$\mathbb{P}(\tau \leq t) \geq 1 - 2\exp\left[-\frac{1}{\kappa t}\left(t - C'\log\frac{M+1}{\Delta+1}\right)^2\right].$$

*Proof.* Let $\beta = \frac{1}{16}$ and $p_\beta = 1 - \frac{7}{8(1-\beta)} = \frac{1}{15}$. Now we use Lemma 4 and 6. Of course for Lemma 4, we assume $t \geq \frac{1}{p_\beta}\left\lceil\frac{\log\frac{M+1}{2\Delta+1}}{\log(1+\beta)}\right\rceil$. We have

$$
\begin{aligned}
\mathbb{P}(\tau \leq t) &= \mathbb{P}(\tau_1 + \tau_2 \leq t) \\
&\geq \mathbb{P}(\tau_1 \leq t/2, \tau_2 \leq t/2) \\
&\geq \mathbb{P}(\tau_1 \leq t/2) + \mathbb{P}(\tau_2 \leq t/2) - 1 \\
&\geq 1 - \exp\left(-\frac{p_\beta^2}{t}\left(t - \frac{2}{p_\beta}\left\lceil\frac{\log\frac{M+1}{2\Delta+1}}{\log(1+\beta)}\right\rceil\right)^2\right) - \frac{1}{2M+1-\Delta}\left(\frac{7}{8}\right)^{t/2} \\
&= 1 - \exp\left(-\frac{1}{15^2 t}\left(t - 30\left\lceil\frac{\log\frac{M+1}{2\Delta+1}}{\log\frac{17}{16}}\right\rceil\right)^2\right) - \frac{1}{2M+1-\Delta}\left(\frac{7}{8}\right)^{t/2}, \quad (7)
\end{aligned}
$$

where the second inequality holds by the union bound. Now take $t \geq \frac{60}{\log\frac{17}{16}}\log\frac{M+1}{2\Delta+1}$, it follows that

$$\exp\left(-\frac{1}{15^2 t}\left(t - 30\left\lceil\frac{\log\frac{M+1}{2\Delta+1}}{\log\frac{17}{16}}\right\rceil\right)^2\right) \leq \exp\left(-\frac{1}{15^2 t}\left(t - \frac{60}{\log\frac{17}{16}}\log\frac{M+1}{\Delta+1}\right)^2\right).$$

Now, consider the second exponential term in Eq. 7. It holds that

$$
\begin{aligned}
\frac{1}{2M+1-\Delta}\left(\frac{7}{8}\right)^{t/2} &= \exp\left[\log\frac{1}{2M+1-\Delta} - \frac{t}{2}\log\frac{8}{7}\right] \\
&\leq \exp\left[\log\frac{M+1}{\Delta+1} - \frac{t}{15}\right] \\
&= \exp\left[-\frac{1}{15}\frac{1}{t - 15\log\frac{M+1}{\Delta+1}}\left(t - 15\log\frac{M+1}{\Delta+1}\right)^2\right].
\end{aligned}
$$

Moreover, for $t \geq 15 \log \frac{M+1}{\Delta+1}$,

$$\exp\left[-\frac{1}{15}\frac{1}{t-15\log\frac{M+1}{\Delta+1}}\left(t-15\log\frac{M+1}{\Delta+1}\right)^2\right] \leq \exp\left[-\frac{1}{15t}\left(t-15\log\frac{M+1}{\Delta+1}\right)^2\right]$$

$$\leq \exp\left[-\frac{1}{15^2 t}\left(t-\frac{60}{\log\frac{17}{16}}\log\frac{M+1}{\Delta+1}\right)^2\right].$$

Altogether, we have that

$$\exp\left[-\frac{p_\beta^2}{t}\left(t-\frac{2}{p_\beta}\left\lceil\frac{\log\frac{M+1}{2\Delta+1}}{\log(1+\beta)}\right\rceil\right)^2\right] + \frac{1}{2M+1-\Delta}\left(\frac{7}{8}\right)^{t/2} \leq 2\exp\left[-\frac{1}{15^2 t}\left(t-\frac{60}{\log\frac{17}{16}}\log\frac{M+1}{\Delta+1}\right)^2\right].$$

The required result now follows by putting $\kappa = 15^2$ and $C' = 60/\log(17/16)$. $\qquad\square$

## B    NOTATION TABLE

| Notation | Description |
|---|---|
| $w, y$ | Scalars (lowercase letters) |
| $\mathbf{v}$ | Vectors (bold lowercase letters) |
| $v_i$ | $i^{\text{th}}$ component of vector $\mathbf{v}$ |
| $\mathbf{M}$ | Matrices (bold uppercase letters) |
| $\mathbf{W} \in \mathbb{R}^{d_1 \times d_2}$ | Matrix with dimensions $d_1 \times d_2$ |
| $\|\mathbf{v}\|$ | $\ell_2$ norm of vector $\mathbf{v}$ |
| $S_\delta := \{-1, -1+\delta, \dots, 1\}$ | Finite grid set, with $\delta = 2^{-k}$, $k \in \mathbb{N}$ |
| $b \in S_\delta$ | Real number $b$ has precision $\delta$ |
| $S_\delta^d$ | $d$-fold Cartesian product of $S_\delta$ |
| $C, C'$ | Positive absolute constants |
| $f : \mathbb{R}^{d_0} \to \mathbb{R}^{d_\ell}$ | $\ell$-layer neural network |
| $\mathbf{W}_i \in \mathbb{R}^{d_i \times d_{i-1}}$ | Weight matrix of $i$-th layer |
| $\sigma(x) = \max(0, x)$ | ReLU activation function |
| $\sigma(\mathbf{x})$ | Component-wise application: $v_i = \sigma(x_i)$ |

Table 2: Summary of notation.

## C    TAXONOMY OF PREVIOUS WORK

Table 1 presents a qualitative comparison with prior work, highlighting the strengths of our approach. It should be noted, however, that the results in Table 1 are subject to varying assumptions, so a direct comparison can be tricky. We now offer a more detailed account of the comparison. First, it should be noted that in Diffenderfer & Kailkhura (2021); Sreenivasan et al. (2022), only the larger network is quantized, and only to the binary case rather than arbitrary precision. Furthermore, in the last column of Table 1, we compare whether the failure probability decays to zero exponentially with respect to overparameterization. Works in which the required overparameterization is already polynomial (as opposed to logarithmic) such as Malach et al. (2020) and Diffenderfer & Kailkhura (2021) are therefore marked with a cross in the last column.

## D    SLTH-QUANTIZATION RESULTS

In this section, we provide the details of the proof of Theorem 3.

**Lemma 7** (Approximating a univariate linear function). *Let $d$ be some positive integer, $\epsilon$ be any real number in $(0, 1)$ and $\delta = \epsilon/2\Delta d$ where $\Delta \in \mathbb{Z}^+$. Consider a randomly initialized neural network $g(\mathbf{x}) = \mathbf{v}^T \sigma(\mathbf{M}\mathbf{x})$ with $\mathbf{x} \in \mathbb{R}^d$ such that $\mathbf{M} \in S_\delta^{C \log\left(\frac{\frac{1}{\delta}+1}{\Delta+1}\right) \times d}$ and $\mathbf{v} \in S_\delta^{C \log\left(\frac{\frac{1}{\delta}+1}{\Delta+1}\right)}$, where each weight is initialized independently from the uniform distribution over $S_\delta$.*

*Let $\hat{g}(\mathbf{x}) = (\mathbf{s} \odot \mathbf{v})^T \sigma((\mathbf{T} \odot \mathbf{M})\mathbf{x})$ be the pruned network for a choice of binary vector $\mathbf{s}$ and matrix $\mathbf{T}$. If $f_{\mathbf{w}}(\mathbf{x}) = \mathbf{w}^T \mathbf{x}, \mathbf{w} \in S_\delta^d$ is the linear function, $n > C \log \frac{\frac{1}{\delta}+1}{\Delta+1}$, then with probability at least*

$$1 - 4d \exp\left(-\frac{1}{\kappa n}\left(n - C' \log \frac{\frac{1}{\delta}+1}{\Delta+1}\right)^2\right),$$

*we have for any $\mathbf{w} \in S_\delta^d$*

$$\exists\, \mathbf{s}, \mathbf{T} : \sup_{\|\mathbf{x}\|_\infty \leq 1} \left|f_{\mathbf{w}}(\mathbf{x}) - \hat{g}(\mathbf{x})\right| < \varepsilon.$$

*Proof.* We will approximate $\mathbf{w}^T \mathbf{x}$ coordinate-wise.

**Step 1: Pre-processing $\mathbf{M}$.** We first begin by pruning $\mathbf{M}$ to create a block-diagonal matrix $\mathbf{M}'$. Specifically, we create $\mathbf{M}'$ by only keeping the following non-zero entries:

$$\mathbf{M}' = \begin{bmatrix} \mathbf{u}_1 & 0 & \cdots & 0 \\ 0 & \mathbf{u}_2 & \cdots & 0 \\ \vdots & \vdots & \ddots & \vdots \\ 0 & 0 & \cdots & \mathbf{u}_d \end{bmatrix}, \quad \text{where } \mathbf{u}_i \in \mathbb{R}^{C \log\left(\frac{\frac{1}{\delta}+1}{\Delta+1}\right)}.$$

We choose the binary matrix $\mathbf{T}$ to be such that $\mathbf{M}' = \mathbf{T} \odot \mathbf{M}$. We also decompose $\mathbf{v}$ and $\mathbf{s}$ as

$$\mathbf{s} = \begin{bmatrix} \mathbf{s}_1 \\ \mathbf{s}_2 \\ \vdots \\ \mathbf{s}_d \end{bmatrix}, \quad \mathbf{v} = \begin{bmatrix} \mathbf{v}_1 \\ \mathbf{v}_2 \\ \vdots \\ \mathbf{v}_d \end{bmatrix}, \quad \text{where } \mathbf{s}_i, \mathbf{v}_i \in \mathbb{R}^{C \log\left(\frac{\frac{1}{\delta}+1}{\Delta+1}\right)}.$$

Using this notation, we can express our network as the following:

$$(\mathbf{s} \odot \mathbf{v})^T \sigma(\mathbf{M}'\mathbf{x}) = \sum_{i=1}^{d} (\mathbf{s}_i \odot \mathbf{v}_i)^T \sigma(\mathbf{u}_i x_i).$$

**Step 2: Pruning $\mathbf{u}$.** Let $n = C \log\left(\frac{\frac{1}{\delta}+1}{\Delta+1}\right)$ and define the event $E_{i,\varepsilon}$ be the following event from Lemma 2:

$$E_i := \left\{ \sup_{w \in S_\delta} \inf_{\mathbf{s}_i \in \{0,1\}^n} \sup_{|x| \leq 1} \left|wx - (\mathbf{v}_i \odot \mathbf{s}_i)^T \sigma(\mathbf{u}_i x)\right| \leq \Delta\delta \right\}.$$

Define the event $E := \bigcap_i E_i$, the intersection of all the events. For each $i$, Lemma 1 shows that event $E_i$ holds with probability at least

$$1 - 4 \exp\left(-\frac{1}{\kappa n}\left(n - C' \log \frac{\frac{1}{\delta}+1}{\Delta+1}\right)^2\right),$$

because the dimension of $\mathbf{v}_i$ and $\mathbf{u}_i$ is at least $C \log\left(\frac{\frac{1}{\delta}+1}{\Delta+1}\right)$. Taking a union bound we get that the event $E$ holds with probability at least

$$1 - 4d \exp\left(-\frac{1}{\kappa n}\left(n - C' \log \frac{\frac{1}{\delta}+1}{\Delta+1}\right)^2\right).$$

On the event $E$, we obtain the following series of inequalities:

$$\sup_{\mathbf{w}\in S_\delta^d} \inf_{\mathbf{s},\mathbf{T}} \sup_{\|\mathbf{x}\|_\infty\leq 1} \left|\mathbf{w}^T\mathbf{x} - (\mathbf{s}\odot\mathbf{v})^T\sigma((\mathbf{T}\odot\mathbf{M})\mathbf{x})\right|$$

$$\leq \sup_{\mathbf{w}\in S_\delta^d} \inf_{\mathbf{s}\in\{0,1\}^{nd}} \sup_{\|\mathbf{x}\|_\infty\leq 1} \left|\mathbf{w}^T\mathbf{x} - (\mathbf{s}\odot\mathbf{v})^T\sigma(\mathbf{M}'\mathbf{x})\right| \quad \text{(Pruning } \mathbf{M} \text{ according to Step 1.)}$$

$$= \sup_{\mathbf{w}\in S_\delta^d} \inf_{\mathbf{s}_1,\dots,\mathbf{s}_d\in\{0,1\}^n} \sup_{\|\mathbf{x}\|_\infty\leq 1} \left|\sum_{i=1}^d w_i x_i - \sum_{i=1}^d (\mathbf{s}_i\odot\mathbf{v}_i)^T\sigma(\mathbf{u}_i x_i)\right| \quad \text{(Using decomposition above)}$$

$$\leq \sup_{\mathbf{w}\in S_\delta^d} \inf_{\mathbf{s}_1,\dots,\mathbf{s}_d\in\{0,1\}^n} \sup_{\|\mathbf{x}\|_\infty\leq 1} \sum_{i=1}^d \left|w_i x_i - (\mathbf{s}_i\odot\mathbf{v}_i)^T\sigma(\mathbf{u}_i x_i)\right|$$

$$= \sum_{i=1}^d \sup_{|w_i|\leq 1} \inf_{\mathbf{s}_i\in\{0,1\}^n} \sup_{|x_i|\leq 1} \left|w_i x_i - (\mathbf{s}_i\odot\mathbf{v}_i)^T\sigma(\mathbf{u}_i x_i)\right|$$

$$\leq \sum_{i=1}^d 2\Delta\delta \quad \text{(By definition of the event } E)$$

$$\leq 2d\Delta\delta.$$

Since $\delta = \varepsilon/2d\Delta$, the result follows. $\qquad\square$

**Lemma 8** (Approximating a layer). *Let $d_1, d_2$ be some positive integers, $\epsilon$ be any real number in $(0,1)$ and $\delta = \epsilon/2\Delta d_1 d_2$ where $\Delta \in \mathbb{Z}^+$. Consider a randomly initialized two-layer neural network $g(x) = \mathbf{N}\,\sigma(\mathbf{M}\mathbf{x})$ with $\mathbf{x} \in \mathbb{R}^{d_1}$ such that $\mathbf{N}$ has dimension*

$$d_2 \times C d_1 \log\left(\frac{\frac{1}{\delta}+1}{\Delta+1}\right)$$

*and $\mathbf{M}$ has dimension*

$$C d_1 \log\left(\frac{\frac{1}{\delta}+1}{\Delta+1}\right) \times d_1,$$

*where each weight is initialized independently from the uniform distribution over $S_\delta$.*

*Let $\widehat{g}(x) = (\mathbf{S}\odot\mathbf{N})^T\sigma\big((\mathbf{T}\odot\mathbf{M})x\big)$ be the pruned network for a choice of pruning matrices $\mathbf{S}$ and $\mathbf{T}$. If $f_{\mathbf{W}}(x) = \mathbf{W}x$ is the linear (single-layered) network, where $\mathbf{W} \in S_\delta^{d_1\times d_2}$, $n > C\log\frac{\frac{1}{\delta}+1}{\Delta+1}$, then with probability at least*

$$1 - 4d_1 d_2 \exp\left(-\frac{1}{\kappa n}\left(n - C'\log\frac{\frac{1}{\delta}+1}{\Delta+1}\right)^2\right),$$

*we have for any $\mathbf{W} \in S_\delta^{d_1\times d_2}$*

$$\exists\, \mathbf{S},\mathbf{T}: \sup_{\mathbf{W}\in S_\delta^{d_1\times d_2}} \sup_{x:\|x\|_\infty\leq 1} \left\|f_{\mathbf{W}}(x) - \widehat{g}(\mathbf{x})\right\| < \varepsilon.$$

*Proof.* Our proof strategy is similar to the proof in Lemma 7.

**Step 1: Pre-processing $\mathbf{M}$.** Similar to Lemma 2, we begin by pruning $\mathbf{M}$ to get a block-diagonal matrix $\mathbf{M}'$:

$$\mathbf{M}' = \begin{bmatrix} \mathbf{u}_1 & 0 & \dots & 0 \\ 0 & \mathbf{u}_2 & \dots & 0 \\ \vdots & \vdots & \ddots & \vdots \\ 0 & 0 & \dots & \mathbf{u}_{d_1} \end{bmatrix}, \qquad \mathbf{u}_i \in \mathbb{R}^{C\log\left(\frac{\frac{1}{\delta}+1}{\Delta+1}\right)}.$$

Thus $\mathbf{T}$ is chosen so that $\mathbf{M}' = \mathbf{T} \odot \mathbf{M}$. We also decompose $\mathbf{N}$ and $\mathbf{S}$ as follows:

$$\mathbf{S} = \begin{bmatrix} \mathbf{s}_{1,1}^T & \cdots & \mathbf{s}_{1,d_1}^T \\ \mathbf{s}_{2,1}^T & \cdots & \mathbf{s}_{2,d_1}^T \\ \vdots & \ddots & \vdots \\ \mathbf{s}_{d_2,1}^T & \cdots & \mathbf{s}_{d_2,d_1}^T \end{bmatrix}, \qquad \mathbf{N} = \begin{bmatrix} \mathbf{v}_{1,1}^T & \cdots & \mathbf{v}_{1,d_1}^T \\ \mathbf{v}_{2,1}^T & \cdots & \mathbf{v}_{2,d_1}^T \\ \vdots & \ddots & \vdots \\ \mathbf{v}_{d_2,1}^T & \cdots & \mathbf{v}_{d_2,d_1}^T \end{bmatrix},$$

where $\mathbf{v}_{i,j},\ \mathbf{u}_i \in \mathbb{R}^{C \log\left(\frac{\frac{1}{\delta}+1}{\Delta+1}\right)}$.

Using this notation we obtain:

$$(\mathbf{S} \odot \mathbf{N})\, \sigma(\mathbf{M}'x) = \begin{bmatrix} \sum_{j=1}^{d_1} (\mathbf{s}_{1,j} \odot \mathbf{v}_{1,j})^T \, \sigma(\mathbf{u}_j x_j) \\ \vdots \\ \sum_{j=1}^{d_1} (\mathbf{s}_{d_2,j} \odot \mathbf{v}_{d_2,j})^T \, \sigma(\mathbf{u}_j x_j) \end{bmatrix}.$$

**Step 2: Pruning N.** Note that $\mathbf{v}_{i,j}$ and $\mathbf{u}_i$ contain i.i.d. random variables from the uniform distribution. Let $n = C \log\left(\frac{\frac{1}{\delta}+1}{\Delta+1}\right)$ and define the event $E_{i,j}$ (from Lemma 1) by

$$E_{i,j} := \left\{ \sup_{w \in S_\delta} \inf_{\mathbf{s}_{i,j} \in \{0,1\}^n} \sup_{x:|x|\leq 1} \left| wx - (\mathbf{v}_{i,j} \odot \mathbf{s}_{i,j})^T \sigma(\mathbf{u}_i x) \right| \leq 2\Delta\delta \right\}.$$

Let $E := \bigcap_{1 \leq i \leq d_2} \bigcap_{1 \leq j \leq d_1} E_{i,j}$ be the intersection of all individual events. By Lemma 1 each event $E_{i,j}$ holds with probability

$$1 - 4\exp\left( -\frac{1}{\kappa n}\left( n - C' \log \frac{\frac{1}{\delta}+1}{\Delta+1} \right)^2 \right),$$

since $\mathbf{u}_i$ and $\mathbf{v}_{i,j}$ have dimension at least $C \log\left(\frac{\frac{1}{\delta}+1}{\Delta+1}\right)$. By a union bound, $E$ holds with probability at least

$$1 - 4d_1 d_2 \exp\left( -\frac{1}{\kappa n}\left( n - C' \log \frac{\frac{1}{\delta}+1}{\Delta+1} \right)^2 \right).$$

On the event $E$ we have the following inequalities:

$$\sup_{\mathbf{W} \in S_\delta^{d_1 \times d_2}} \inf_{\mathbf{S},\mathbf{T}} \sup_{\|x\|_\infty \leq 1} \left\| \mathbf{W}x - (\mathbf{S} \odot \mathbf{N})^T \sigma\big((\mathbf{T} \odot \mathbf{M})x\big) \right\|$$

$$\leq \sup_{\mathbf{W} \in S_\delta^{d_1 \times d_2}} \inf_{\mathbf{S}} \sup_{\|x\|_\infty \leq 1} \left\| \mathbf{W}x - (\mathbf{S} \odot \mathbf{N})^T \sigma(\mathbf{M}'x) \right\| \qquad \text{(Pruning } \mathbf{M} \text{ as in Step 1)}$$

$$\leq \sup_{\mathbf{W} \in S_\delta^{d_1 \times d_2}} \inf_{\mathbf{s}_{i,j} \in \{0,1\}^n} \sup_{\|x\|_\infty \leq 1} \sum_{i=1}^{d_2} \left| \sum_{j=1}^{d_1} w_{i,j} x_j - \sum_{j=1}^{d_1} (\mathbf{s}_{i,j} \odot \mathbf{v}_{i,j})^T \sigma(\mathbf{u}_j x_j) \right|$$

$$\leq \sup_{|w_{i,j}| \leq 1} \inf_{\mathbf{s}_{i,j} \in \{0,1\}^n} \sup_{|x_j| \leq 1} \sum_{i=1}^{d_2} \sum_{j=1}^{d_1} \left| w_{i,j} x_j - (\mathbf{s}_{i,j} \odot \mathbf{v}_{i,j})^T \sigma(\mathbf{u}_i x_j) \right|$$

$$\leq \sup_{|w_{i,j}| \leq 1} \inf_{\mathbf{s}_{i,j} \in \{0,1\}^n} \sum_{i=1}^{d_2} \sum_{j=1}^{d_1} \sup_{|x_j| \leq 1} \left| w_{i,j} x_j - (\mathbf{s}_{i,j} \odot \mathbf{v}_{i,j})^T \sigma(\mathbf{u}_i x_j) \right|$$

$$= \sum_{i=1}^{d_2} \sum_{j=1}^{d_1} \sup_{|w| \leq 1} \inf_{\mathbf{s} \in \{0,1\}^n} \sup_{|x| \leq 1} \left| wx - (\mathbf{s} \odot \mathbf{v}_{i,j})^T \sigma(\mathbf{u}_i x) \right|$$

$$\leq 2\Delta\delta d_1 d_2.$$

Since $\delta = \varepsilon / 2d_1 d_2 \Delta$, the result follows. $\qquad\square$

We now use Lemma 8 to prove Theorem 3.

*Proof of Theorem 3.* Let $\mathbf{x}_i$ be the input to the $i$-th layer of $f_{(\mathbf{W}_1,\dots,\mathbf{W}_\ell)}(\mathbf{x})$. Thus,

1. $\mathbf{x}_1 = \mathbf{x}$,
2. for $1 \leq i \leq \ell - 1$, $\mathbf{x}_{i+1} = \sigma(\mathbf{W}_i \mathbf{x}_i)$.

Thus $f_{(\mathbf{W}_l,\dots,\mathbf{W}_1)}(\mathbf{x}) = \mathbf{W}_l \mathbf{x}_l$.

For the $i$-th layer weights $\mathbf{W}_i$, let $\mathbf{S}_{2i}$ and $\mathbf{S}_{2i-1}$ be the binary matrices that achieve the guarantee in Lemma 8. Lemma 8 states that with probability

$$1 - 4d_1 d_2 \exp\left(-\frac{1}{\kappa n}\left(n - C' \log \frac{\frac{1}{\delta} + 1}{\Delta + 1}\right)^2\right),$$

the $\delta$ is chosen such that the following event holds:

$$\sup_{\mathbf{W} \in S_\delta^{d_{i+1} \times d_i}} \exists\, \mathbf{S}_{2i}, \mathbf{S}_{2i-1} : \sup_{\mathbf{x}:\|\mathbf{x}\| \leq 1} \left\|\mathbf{W}_i \mathbf{x} - (\mathbf{M}_{2i} \odot \mathbf{S}_{2i})\, \sigma\big((\mathbf{S}_{2i} \odot \mathbf{M}_{2i-1})\mathbf{x}\big)\right\| < \frac{\varepsilon}{2\ell}. \quad (8)$$

As ReLU is 1-Lipschitz, the above event implies the following:

$$\sup_{\mathbf{W} \in S_\delta^{d_{i+1} \times d_i}} \exists\, \mathbf{S}_{2i}, \mathbf{S}_{2i-1} : \sup_{\mathbf{x}:\|\mathbf{x}\| \leq 1} \left\|\sigma(\mathbf{W}_i \mathbf{x}) - \sigma\big((\mathbf{M}_{2i} \odot \mathbf{S}_{2i})\, \sigma\big((\mathbf{M}_{2i-1} \odot \mathbf{S}_{2i-1})\mathbf{x}\big)\big)\right\| < \frac{\varepsilon}{2\ell}.$$
$$(9)$$

Taking a union bound, we get that with probability

$$1 - \sum_{i=0}^{\ell-1} 4d_i d_{i+1} \exp\left(-\frac{1}{\kappa n}\left(n - C' \log \frac{\frac{1}{\delta} + 1}{\Delta + 1}\right)^2\right), \quad (10)$$

the above inequalities hold for every layer simultaneously. For the remainder of the proof, we will assume that this event holds. For any fixed function $f$, let $g_f = g_{(\mathbf{W}_1,\dots,\mathbf{W}_\ell)}$ be the pruned network constructed layer-wise, by pruning with binary matrices satisfying the above conditions, and let these pruned matrices be $\mathbf{M}'_i$. Let $\mathbf{x}'_i$ be the input to the $(2i-1)$-th layer of $g_f$. We note that $\mathbf{x}'_i$ satisfies the following recurrent relations:

1. $\mathbf{x}'_1 = \mathbf{x}$,
2. for $1 \leq i \leq \ell - 1$, $\mathbf{x}'_{i+1} = \sigma\big(\mathbf{M}'_{2i}\, \sigma(\mathbf{M}'_{2i-1}\mathbf{x}'_i)\big)$.

Because the input $\mathbf{x}$ has $\|\mathbf{x}\| \leq 1$, Equation 9 also states that $\|\mathbf{x}'_i\| \leq \left(1 + \frac{\varepsilon}{2\ell}\right)^{i-1}$. To see this, note that we use Equation 9 to get for $1 \leq i \leq l - 1$:

$$\|\sigma(\mathbf{W}_i \mathbf{x}'_i) - \mathbf{x}'_{i+1}\| \leq \|\mathbf{x}'_i\| \frac{\varepsilon}{2\ell},$$

which implies

$$\|\mathbf{x}'_{i+1}\| \leq \|\mathbf{x}'_i\| \frac{\varepsilon}{2\ell} + \|\sigma(\mathbf{W}_i \mathbf{x}'_i)\| \leq \|\mathbf{x}'_i\| \frac{\varepsilon}{2\ell} + \|\mathbf{W}_i \mathbf{x}'_i\| \leq \|\mathbf{x}'_i\| \frac{\varepsilon}{2\ell} + \|\mathbf{x}'_i\|.$$

Applying this inequality recursively yields the claim that for $1 \leq i \leq l - 1$,

$$\|\mathbf{x}'_i\| \leq \left(1 + \frac{\varepsilon}{2\ell}\right)^{i-1}.$$

Using this, we can bound the error between $\mathbf{x}_i$ and $\mathbf{x}_i'$. For $1 \le i \le l-1$, we have

$$
\begin{aligned}
\|\mathbf{x}_{i+1} - \mathbf{x}_{i+1}'\| &= \left\|\sigma(\mathbf{W}_i\mathbf{x}_i) - \sigma(\mathbf{M}_{2i}'\sigma(\mathbf{M}_{2i-1}'\mathbf{x}_i'))\right\| \\
&\le \left\|\sigma(\mathbf{W}_i\mathbf{x}_i) - \sigma(\mathbf{W}_i\mathbf{x}_i')\right\| + \left\|\sigma(\mathbf{W}_i\mathbf{x}_i') - \sigma(\mathbf{M}_{2i}'\sigma(\mathbf{M}_{2i-1}'\mathbf{x}_i'))\right\| \\
&\le \|\mathbf{x}_i - \mathbf{x}_i'\| + \|\mathbf{W}_i\mathbf{x}_i' - \mathbf{M}_{2i}'\sigma(\mathbf{M}_{2i-1}'\mathbf{x}_i')\| \\
&< \|\mathbf{x}_i - \mathbf{x}_i'\| + \left(1 + \frac{\varepsilon}{2\ell}\right)^{i-1}\frac{\varepsilon}{2\ell},
\end{aligned}
$$

where we use Equation 8. Unrolling this we get

$$
\|\mathbf{x}_\ell - \mathbf{x}_\ell'\| \le \sum_{i=1}^{\ell-1}\left(1 + \frac{\varepsilon}{2\ell}\right)^{i-1}\frac{\varepsilon}{2\ell}.
$$

Finally using the inequality above, we get that with probability at least $1 - \varepsilon$,

$$
\begin{aligned}
\|f(\mathbf{W}_\ell, \ldots, \mathbf{W}_1)(\mathbf{x}) - g(\mathbf{W}_\ell, \ldots, \mathbf{W}_1)(\mathbf{x})\| &= \left\|\mathbf{W}_\ell\mathbf{x}_\ell - \mathbf{M}_{2\ell}'\sigma(\mathbf{M}_{2\ell-1}'\mathbf{x}_l')\right\| \\
&\le \|\mathbf{W}_\ell\mathbf{x}_\ell - \mathbf{W}_\ell\mathbf{x}_\ell'\| + \|\mathbf{W}_\ell\mathbf{x}_\ell' - \mathbf{M}_{2\ell}'\sigma(\mathbf{M}_{2\ell-1}'\mathbf{x}_\ell')\| \\
&\le \|\mathbf{x}_\ell - \mathbf{x}_\ell'\| + \left(1 + \frac{\varepsilon}{2\ell}\right)^{\ell-1}\frac{\varepsilon}{2\ell} \\
&\le \left(\sum_{i=1}^{\ell-1}\left(1 + \frac{\varepsilon}{2\ell}\right)^{i-1}\frac{\varepsilon}{2\ell}\right) + \left(1 + \frac{\varepsilon}{2\ell}\right)^{\ell-1}\frac{\varepsilon}{2l} \\
&\le \sum_{i=1}^{\ell}\left(1 + \frac{\varepsilon}{2\ell}\right)^{i-1}\frac{\varepsilon}{2\ell} \\
&= \left(1 + \frac{\varepsilon}{2\ell}\right)^{\ell} - 1 \\
&< e^{\varepsilon/2} - 1 \\
&< \varepsilon. \quad \text{(since } \varepsilon < 1\text{)}
\end{aligned}
$$

Now simply using the definition of $n$ in Equation 10 gives the required result. $\qquad\square$

## E EXPERIMENTS

In this section, we experimentally validate our results for the subset sum problem. To illustrate, we consider a set of integers $X_1, X_2, \ldots, X_n$ sampled uniformly from the set $\{-M, \ldots, M\}$. We focus on the case of exact representation, i.e., $\Delta = 0$, which corresponds to the hardest scenario (allowing a tolerance generally makes the problem easier). We solve the subset sum problem for a target value $z = 42$ as an example. For given values of $M$ and $n$, we randomly sample $X_1, X_2, \ldots, X_n$ and determine whether the subset sum problem has a solution. This procedure is repeated multiple times to estimate the probability that a randomly sampled instance is solvable. We then perform this experiment for several choices of $M$ and $n$. Figure 1 shows the probability of solving the subset sum problem as a function of $n$ for various values of $M$. According to Theorem 2, the probability of successfully solving the subset sum problem should be close to 1 whenever $n > C\log(M + 1)$. By analyzing the plots, one can estimate that $C \approx 1.4$ (See Figure 1).

According to Theorem 3, the task of approximating a given target network by pruning a large network is equivalent to solving a collection of independent random subset sum problems. Previous work on SLTH (Cunha et al., 2023) explicitly construct this sparse network for a given target network by solving many of these independent random subset sum problems. As this procedure is straightforward, we do not elaborate on it here.

We now estimate the running time required to identify a subnetwork. Consider, for example, ResNet-50, which has 23 million ($2.3 \times 10^7$) parameters, and let each weight be stored using an FP8 (8-bit

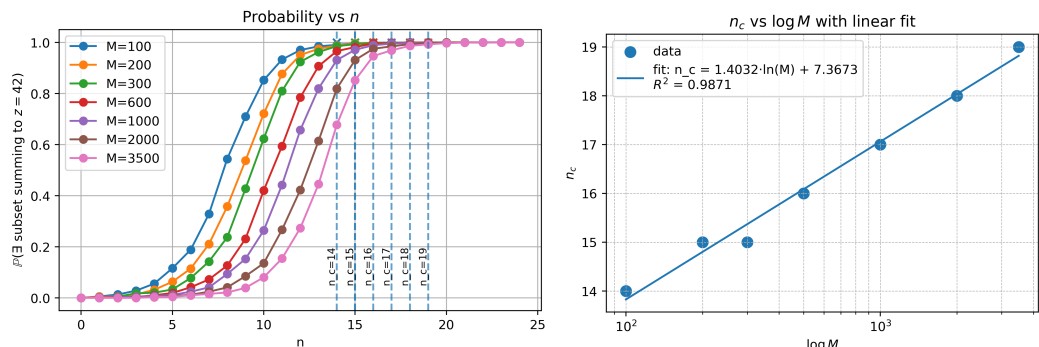

Figure 1: Left: Probability of solving a Random subset sum problem as a function of $n$ for several values of $M$ for target $z = 42$. The probability was computed over 20000 trials. $n_c$ is the value of $n$ after which probability is greater than 0.99. Right: $n_c$ vs $\log M$ with a linear fit to determine the constant $C$.

floating-point) format. In this setting, $M = 2^7$, and thus $n > C\log(2^7)$ is sufficient according to our theory. From our experiments, the average time to solve a single random subset sum instance with $n = 50$ and $M = 128$ is approximately $7 \times 10^{-5}$ seconds (averaged over $10^5$ trials). Consequently, solving $2.3 \times 10^7$ such instances would require roughly 27 minutes, without any parallelization.

## F  REJECTION SAMPLING

Let $Y$ be a discrete random variable taking values in the finite set

$$\mathcal{Y} = \{y_1, y_2, \ldots, y_M\}.$$

Let its probability mass function (pmf) be $p_i = \mathbb{P}(Y = y_i)$ for all $i \in \{1, \ldots, M\}$ and assume $p_i > 0$ for all $i$. Define

$$p_{\min} := \min_{1 \le i \le M} p_i > 0.$$

Let $U$ be the uniform distribution on $\mathcal{Y}$, i.e.

$$\mathbb{P}(U = y_i) = \frac{1}{M}.$$

Define

$$\alpha := M p_{\min},$$

which satisfies $\alpha \le 1$. Now define another pmf $g = (g_1, \ldots, g_M)$ by

$$g_i := \frac{p_i - \frac{\alpha}{M}}{1 - \alpha} = \frac{p_i - p_{\min}}{1 - M p_{\min}}, \qquad i = 1, \ldots, M.$$

Let $B \sim \mathrm{Bernoulli}(\alpha)$, independent of both $U$ and a random variable $G$ with pmf $g$. Then

$$Y \overset{d}{=} \begin{cases} U, & B = 1, \\ G, & B = 0. \end{cases}$$

Equivalently, the distribution of $Y$ satisfies

$$\mathbb{P}(Y = y_i) = \alpha \cdot \frac{1}{M} + (1 - \alpha)g_i, \qquad \forall i.$$

Thus the distribution of $Y$ can be expressed as

$$Y \overset{d}{=} B\,U + (1 - B)\,G,$$

where $U \sim \mathrm{Unif}(\mathcal{Y})$, $G \sim g$, and $B \sim \mathrm{Bernoulli}(\alpha)$. Consider $n$ samples $X_1, X_2, \ldots, X_n$ from this distribution. We now show that if $n$ is large enough, a constant fraction of $X_i$'s are uniformly distributed.

**Theorem 5.** *For $n$ large enough, a constant fraction $\alpha/2 = (Mp_{\min})/2$ of $X_i$'s are uniformly distributed with probability at least $1 - \exp(-n(Mp_{\min})^2/2)$.*

*Proof.* Let $B_1, \ldots, B_n$ be the i.i.d. Bernoulli$(\alpha)$ indicators corresponding to the $n$ samples, and define

$$S := \sum_{i=1}^{n} B_i, \qquad \mathbb{E}[S] = n\alpha = nMp_{\min}.$$

By Hoeffding's inequality, for $t = n\alpha/2$,

$$\mathbb{P}\left(S \leq \frac{n\alpha}{2}\right) = \mathbb{P}\left(S - \mathbb{E}[S] \leq -t\right) \leq \exp\left(-\frac{2t^2}{n}\right) = \exp\left(-\frac{n\alpha^2}{2}\right) = \exp\left(-\frac{n(Mp_{\min})^2}{2}\right).$$

Thus, with probability at least $1 - \exp(-n(Mp_{\min})^2/2)$,

$$S \geq \frac{n\alpha}{2} = \frac{nMp_{\min}}{2}.$$

Hence at least a constant fraction $\alpha/2 = (Mp_{\min})/2$ of the samples come from the uniform component with high probability. $\square$

## G  LLM USAGE

We emphasize that Large Language Models (LLMs) were used solely to improve the clarity and readability of the manuscript. Specifically, we employed LLM assistance to refine sentence structure, grammar, and presentation of ideas without altering the technical content, analysis, or conclusions. All conceptual contributions, theoretical developments, and experimental results presented in this work are entirely original and conducted by the authors.

