# OpenReview forum: "A Unified Framework for Quantized and Continuous Strong Lottery Tickets"
_ICLR.cc/2026/Conference — Submitted to ICLR 2026_

### Official Review · Reviewer_bZyu · 2025-10-27

**Soundness:** 2
**Presentation:** 3
**Contribution:** 3
**Rating:** 4
**Confidence:** 2

**Summary:**

This paper presents a unified theoretical framework for the Strong Lottery Ticket Hypothesis (SLTH) that bridges the gap between continuous and quantized settings. The authors extend the analysis of the Random Subset Sum Problem (RSSP) to the discrete regime, obtaining exponentially decaying failure probabilities in the quantized case—an exponential improvement over prior polynomial bounds.

**Strengths:**

I am not an expert in this specific research area, so I cannot confidently assess the novelty of the theoretical contribution, but the results appear technically interesting.

**Weaknesses:**

I would propose for further clarification on questions below:

**Questions:**

-As I am not an expert in this field, I am curious why the existence of such an optimal sub-network is particularly interesting. Could you elaborate on the practical implications of this theoretical result?
I am also wondering how one could actually find or reach this optimal sub-network in practice. Since it seems like a combinatorial problem.
Is there any theoretical insight into whether standard pruning or quantization heuristics could approximate the theoretically optimal sub-network?

-I am also a bit confused about the notation $n$ in the paper. Could you clarify what it represents in this context? In Theorem 3 it is stated that mask matrices should be of the order of $n$? I would propose the authors for clarification.

-About the exponential decay of the failure probability in Theorem 2:  Could the authors comment on whether this rate depends sensitively on the distributional assumptions of the random weights?

---

> ### Author Response · Authors · 2025-11-15
>
> We thank the reviewer for their precious feedback.
>
> Sparsification of neural networks has long been a central problem in machine learning. The Strong Lottery Ticket Hypothesis (SLTH) emerged from the empirical observation (see Related Works for citations) that simply sparsifying a network can often yield highly performant subnetworks. In particular, researchers found that one can identify masks which, when applied to neural networks, result in models that perform very well on given tasks. Theoretical results on SLTH aim to formally establish the existence of such sparse subnetworks, thereby validating these empirical findings. Also, for a discussion on how to find these optimal subnetworks, we request that the reviewer have a look at our official comment, where we address the question of experiments. It is indeed a challenge for future work to understand the subnetworks found by standard pruning or quantization heuristics.
>
> In Theorem 3, $n$ (the overparameterization factor) denotes the ratio between the width of the larger network and that of the target network. In essence, it tells us that, given any target network, one requires a network that is $n$ times wider, from which pruning can yield an approximation of the target network.
>
> The overparameterization factor will change by a constant depending on the initialization distribution of the weights (of the larger network), but the decay rate of the failure probability will remain the same. Everything remains the same if we change the initialization distribution of the target network.
>
> Please let us know if our response satisfactorily addresses all your concerns regarding our paper. If not, we would be happy to provide further clarifications.

---

> > ### Comment · Reviewer_bZyu · 2025-11-26
> >
> > Thanks to the authors for their response. While studying the existence of a sparse sub-network within an overparameterized network that can approximate a target function is valuable, I remain unconvinced that such a sub-network can realistically be reached in practice by manipulating a large, overparameterized model.  For this reason, I would keep my score as it is and leave the decision for those who are more expert in this field.

---

### Official Review · Reviewer_wd7Q · 2025-10-27

**Soundness:** 2
**Presentation:** 2
**Contribution:** 1
**Rating:** 2
**Confidence:** 4

**Summary:**

This paper studies the theoretical bounds of strong lottery ticket hypothesis under the condition that the randomly initialized neural network's weights are quantized. To start, the paper provided an extension to the random subset sum problem by letting the target value and the randomly generated value taken from a discrete set. In this problem, the paper discretizes the proof from prior work and derives a sample complexity that is logarithmic with respect to the target accuracy. Moreover, the paper applied this bound to the strong lottery ticket hypothesis, and leveraged the standard proof technique to show the overparameterization required for a good approximation of a target network with discrete weights.

**Strengths:**

1. The paper considers the problem of pruning and quantization, which is an important topic by it self.
2. The paper extended the proof of random subset sum problem to a discrete setting.
3. The paper applied the result of the discrete random subset sum to the strong lottery ticket hypothesis and derived corresponding overparameterization requirement to achieve a given approximation error.

**Weaknesses:**

1. The setting considered by the paper is a little bit confusing. In particular, the paper considers approximating a target network with discrete value, which would intuitively be a simpler task than approximating a continuous value (e.g. approximating any value in $[-M, M]$ is intuitively harder than approximating values in $\{-M, -M+1, \dots, M-1, M\}$). However, the paper claims that approximating the discrete value is more general. Moreover, the paper considers initializing the student network (to be pruned) from discrete distribution. Ideally a more common approach is to initialize from continuous distribution, do the pruning to approximate the target network, and then do quantization. The paper is not clear about why they did not choose the latter approach.
2. The setting of the discrete random subset sum problem is simple. In particular, the paper only considers approximating $Z$ from $\{-M, -M+1, \dots, M-1, M\}$ using subset sum of $X_i$'s sampled from the same candidate set. It is not clear how the derived result is better than simply consider sampling $n$ copies of $X_i$'s, and computing the probability that at least one $X_i$ is $Z$ (which would have probability $1 - (\frac{2M}{2M+1})^n$). This simple bound gives an exact approximation, and logarithmic dependency on the probability. Compared with this bound, the bound provided in the paper seems to be worth since the success probability does not decay exponentially with respect to the number of samples.
. 3The paper lacks necessary experiments to validate the theoretical results.

**Questions:**

None

---

> ### Author Response · Authors · 2025-11-15
>
> We thank the reviewer for their precious feedback.
>
> 1.) We would like to point out that computers always represent numbers with finite precision. In this context, it is natural and interesting to ask how the degree of overparameterization required by SLTH depends on the number of bits used to represent each weight. We refer the reviewer to Kumar and Natale, 2025 for further motivation on this problem. It is indeed possible to begin with higher-precision weights, prune the network, and subsequently discard unnecessary bits through quantization. This idea was already considered theoretically by Kumar and Natale, 2025 and continues to hold in our framework.
>
> 2.) This is a very good question. In fact, the same reasoning applies to the continuous setting as well: one can divide the interval into patches of size $\epsilon$ and apply the same argument. This would seemingly imply that all the classic results on RSS (Lueker, 1998) are redundant - which is fortunately not the case. The key difference lies in how many samples are required to achieve a small failure probability. In the classic results, this number scales as $\log(1/\epsilon)$, whereas in this argument it scales as $\log(\epsilon) / \log(1 - \epsilon/2)$. The latter grows much faster as $\epsilon \to 0$, infact it scales like $-2\log \epsilon/\epsilon$ around $\epsilon=0$. The argument is the same in the discrete setting. The number of samples required is central to all SLTH analyses, as it directly corresponds to the degree of overparameterization - and our goal is to minimize overparameterization. The first result on SLTH (Malach et al., 2020) used the following approach: approximate the target by ensuring that, within a set of weights, there exists at least one weight that matches the target weight exactly. Several improvements have been developed since then.
>
> 3.) We acknowledge that the paper focuses exclusively on theoretical analysis. We respectfully refer the reviewer to our official comment, where we explain our rationale for not including empirical experiments in the original submission. We have now added an experimental component to the paper; please see the official comment for details.
>
> Please let us know if our response satisfactorily addresses all your concerns regarding our paper. If not, we would be happy to provide further clarifications.

---

> > ### Comment · Reviewer_wd7Q · 2025-11-26
> >
> > Thank you for your rebuttal. I have several questions not cleared.
> >
> > 1. I understand the importance of studying things under finite  precision, and its relationship to overparameterization. My main question is why the paper claims that approximating discrete numbers is a more challenging task than approximating continuous numbers, as the latter seems more difficult to me.
> >
> > 2. How would the result in this paper compare with (in terms of overparameterization requirement) Kumar and Natale, 2025, where their approach is to first approximate continuous weights, and then to discard unnecessary bits?
> >
> > 3. Regarding the sample complexity of the classical approach, if we apply $(1 - \frac{1}{m})^n \leq e^{-\frac{n}{m}}$, then the complexity becomes $n \geq m \log 1 / \epsilon$, where $\epsilon$ is the failing probability. Could you elaborate on why the result in Theorem 2 gives a log dependency on the failing probability and how it compares with this bound?

---

> > > ### Author Response · Authors · 2025-11-27
> > >
> > > We thank the reviewer for further clarifying their questions.
> > >
> > > 1.)  To be precise, when we say that approximating discrete numbers is more challenging than approximating continuous ones, we do not mean that in practice it is harder to find subsets in the discrete setting. Rather, we mean that it is more difficult to establish theoretical guarantees in the discrete case. This is often the case in many areas of mathematics, where the continuous setting allows tools that are not available in the discrete setting (a common approach is, indeed, to approximate the discrete setting with a continuous one). This is also reflected in the prior literature for our problem: nearly all existing SLTH results are proved in the continuous setting, precisely because the corresponding random subset-sum results which dates back to Lueker 1998 are only continuous. The only result in the discrete setting that is even roughly comparable to its continuous analogue is due to Kumar and Natale (2025). Notably, their proof relies on sophisticated machinery from the theory of phase transitions in the Number Partitioning problem. In contrast, our work matches the quality of the continuous-setting results—both in terms of required overparameterization and failure probability—without appealing to very advanced analytical tools, relying instead on mathematical arguments that are relatively more elementary. Crucially, the previous continuous results emerge as a special case of ours in the limit where the quantization goes to zero. Likewise, by letting the approximation error vanish, we obtain a guarantee that strictly improves upon that of Kumar and Natale in terms of failure probability.
> > >
> > > 2.) Our required overparameterization matches that of Kumar and Natale (2025), while our failure probability is exponentially smaller, addressing a key limitation of their work. Please also refer to point 1 above, regarding the comparison with their techniques and the generality of our result.
> > >
> > > 3.) This is indeed the heart of the mathematical difficulty of the problem. One can leverage a bound such as $\left(1-\frac{1}{m}\right)^n \leq e^{-\frac{n}{m}}$ to obtain a small failure probability, but the sample complexity would be much worse in that case: for the same failure probability, it would result in a sample size which is linear in $m$, while our analysis yields a sample size of order $\log m$, which is exponentially smaller. This is the true value of Theorem 2. We acknowledge that pointing this out makes understanding the contribution of the paper much easier for readers who are not very familiar with previous works on SLTH, and we are adding this key remark to the introduction of the revised manuscript.

---

### Official Review · Reviewer_T44f · 2025-10-29

**Soundness:** 2
**Presentation:** 2
**Contribution:** 3
**Rating:** 4
**Confidence:** 3

**Summary:**

Summary:

This paper presents a unified theoretical framework for the Strong Lottery Ticket Hypothesis (SLTH) that encompasses both continuous (arbitrary-precision) and quantized (finite-precision) neural networks. The SLTH posits that large, randomly initialized networks contain sparse subnetworks ("tickets") that can achieve high performance without any training. A key theoretical tool for studying SLTH is the Random Subset Sum Problem (RSSP). The authors identify a gap in existing literature: theoretical guarantees for quantized SLTH provided only inverse-polynomial decay for failure probability and only applied to exact representations, which could be slightly loosen as approximate cases for tighter bound of failure probability.

The core of this work is a new, sharp analysis of the RSSP in the discrete setting666. This analysis (Theorem 2) establishes an exponentially small failure probability for finding a $\Delta$-approximation, which is a significant improvement. By applying this new discrete RSSP result, the authors derive tight SLTH guarantees for quantized networks (Theorem 3). Their framework successfully unifies the field by simultaneously handling approximation errors ($\epsilon$) and rounding/quantization errors ($\delta$). They show that previous results for continuous-approximate tickets and quantized-exact tickets emerge as natural limiting cases of their single, unified model.


The primary contributions of this paper are:

1. A Unified Framework: It develops the first theoretical framework that unifies the study of SLTH for both continuous (approximate) and quantized (exact and approximate) networks.

2. Exponential Probability Bounds: The paper's analysis of the discrete RSSP (Theorem 2) yields an exponentially decaying failure probability bound, as shown in the equation:$P(\tau \le t) \ge 1 - 2 \exp\left[-\frac{1}{\kappa t}\left(t - C \log \frac{M+1}{\Delta+1}\right)^2\right]$ This is an exponential improvement over previous inverse-polynomial bounds for the quantized SLTH.

3. Tight Quantized SLTH Guarantees: By applying their new RSSP result, the authors prove (Theorem 3) that a sufficiently overparameterized, randomly initialized network with quantized weights (from $S_\delta$) can $\epsilon$-approximate any target network with exponentially high probability.

**Strengths:**

* Theoretical Significance: The unification of the continuous and quantized SLTH literature is a major theoretical achievement.
* Improved Bounds: The exponential improvement of the failure probability bound (from inverse-polynomial to exponential) is a very strong result. It tightens the theoretical guarantees significantly, bringing the quantized case in line with the continuous one.
* Novel Analysis: The core of the paper, the new analysis for the discrete RSSP, is a valuable combinatorial contribution in its own right and serves as the engine for the paper's other results.
* Generality: The framework's ability to simultaneously manage approximation error ($\epsilon$) and quantization error ($\delta$) makes it powerful and more reflective of practical scenarios where precision is finite.

**Weaknesses:**

* No Empirical Validation: The paper appears to be purely theoretical. It does not present any empirical experiments to validate the new, tighter bounds or to demonstrate how the required overparameterization $\mathcal{O}(d \log(1/\delta))$ behaves in practice.
* Assumptions: The analysis relies on weights being sampled uniformly from the discrete set $S_\delta$. The authors state this can be relaxed using a "standard rejection sampling argument", but this critical step is not fully elaborated upon in the main text.
* Proof Deferral: As is common, the full proof of the core technical result (Theorem 2) is deferred to the appendix. The main paper relies on a proof sketch, since I don't have time to read supplementary, this step makes me unsure of the soundness of this major proof, the sketch looks reasonable though.
* Existential (what) vs. Efficiently Search (how): The paper proves the existence of quantized strong lottery tickets but does not (and does not claim to) provide new, efficient algorithms for finding them. The problem of efficiently finding the optimal subnetwork remains a challenge.

**Questions:**

1. Your theoretical bounds are an exponential improvement. How do these new bounds compare to empirical observations? Can you design an experiment to show that the failure probability of finding a quantized ticket does indeed scale exponentially with overparameterization, as your theory predicts?

2. In the proof of Lemma 1, you use a rejection sampling argument to handle the non-uniform distribution of $Z_i = b_i a_i^+$. Could you elaborate on how this step affects the constants in your final overparameterization bound $n \ge C \log \frac{1/\delta + 1}{\Delta+1}$?

3. How precisely does your framework recover the existing continuous SLTH bounds (e.g., from Pensia et al.) in the limit as the quantization step $\delta \to 0$?

4. The analysis is for fully-connected networks with ReLU activations. What are the primary obstacles to extending this unified (quantized + approximate) framework to more complex architectures like Transformers or ConvNets?

---

> ### Author Response · Authors · 2025-11-15
>
> We thank the reviewer for their precious feedback.
>
> ## Response to Weaknesses
>
> Empirical validation: We acknowledge that the paper focuses exclusively on theoretical analysis. We respectfully refer the reviewer to our official comment, where we explain our rationale for not including empirical experiments in the original submission. We have now added an experimental component to the paper; please see the official comment for details.
>
> Assumptions: We acknowledge the missing clarification on the rejection sampling argument. To address this, we have added a new appendix (Appendix F) providing a rigorous justification.
>
> Proof Deferral: Regarding the proof structure, unfortunately, due to space constraints, we had to defer the full proof of Theorem 2 to the appendix. We welcome any suggestions on whether particular proof sketches should include more details, even at the expense of shortening other proofs.
>
> Existential (what) vs. Efficiently Search (how):  As for efficiently finding the optimal sparse subnetworks, several experimental works have addressed this challenge—please see the section on related works and our official comment for details.
>
>
> ## Response to Questions
>
> 1.) Please refer to our official comment, where the question of experiments is discussed in detail.
>
> 2.) The rejection sampling argument will indeed increase the constant factor in the overparameterization bound. See the newly added Appendix F for a detailed explanation.
>
> 3.) Our framework naturally recovers the continuous SLTH bounds (e.g., Pensia et al.) in the limit as the quantization step tends to zero. This correspondence is discussed in Section 4.1.
>
> 4.) SLTH has already been established for CNNs (Cunha et al., 2023) and Transformers (Otsuka et al., 2025). Our goal in this work was to unify the quantized and continuous settings. While our framework can in principle be extended to other architectures, we leave this as an avenue for future work.
>
> Please let us know if our response satisfactorily addresses all your concerns regarding our paper. If not, we would be happy to provide further clarifications.

---

> > ### Comment · Reviewer_T44f · 2025-11-27
> >
> > Thanks for authors for replying my concerns. However, I still have one question remaining:
> > 1. For explanation of why not including empirial validation, authors mention that they "chose not to include such experiments in the paper because their success is essentially guaranteed: each edge of the target network corresponds to an independent RSS instance, and these instances are solved independently of one another.". However, we can only say an over-parameterized network can be pruned the any target network we want, but how they influence each other or keep independency in gradient descent updating is still not clear, and the definition of success is not clear as well. Is the success for training success, test success or the success of an imitation to the performance of a quantized CNN?

---

> > > ### Author Response · Authors · 2025-11-27
> > >
> > > Thank you to the reviewer for the question.
> > >
> > > The definition of success is that of an imitation to the performance of a quantized CNN, and the mathematical proof shows that they keep independency.
> > >
> > > To provide more details, let us state the SLTH problem again. We are given two neural networks:
> > >
> > > 1. The target network.
> > > This can be any arbitrary neural network with a fixed set of weights. You can think of it as a model that has already been trained for some task; its parameters are set and cannot be modified.
> > >
> > > 2. The larger network.
> > > This is a second network, strictly larger than the target network, whose weights are randomly initialized and then fixed.
> > >
> > > The SLTH question is:
> > > > Can we prune the larger network so that, for every input, its output matches that of the target network?
> > >
> > > Importantly, we are not allowed to change any weights in either network. For the target network, the weights are fixed by assumption; for the larger network, we are only permitted to remove weights through pruning—no training or fine-tuning is allowed.
> > >
> > > Essentially all results proving the SLTH under a variety of condition, including our present contribution, satisfying this matching condition reduces to solving a collection of independent Random Subset Sum Problems (RSSPs). These results show that, in a rigorous mathematical sense, all these RSSPs **can be solved independently**, and that their solution means that the larger network can be pruned in such a way that it **approximates the target network’s behavior**.
> > >
> > > Should any aspect of our contribution remain unclear, we welcome any further request for clarification.

---

> ### Comment · Reviewer_T44f · 2025-11-28
>
> Thanks authors for explanation, I get clear now. However, I am more interested about the statement "all these RSSPs can be solved independently", like what this statement brings to the ML research. In other words, what this independency in SLTH brings to the general ML training and test?

---

> ### Author Response · Authors · 2025-11-28
>
> Thank you to the reviewer for the question.
>
> We certainly agree with the reviewer that the independence of the approximation of the edges is a very interesting property of the SLTH construction, and it was indeed the key idea that Malach et al. and Pensia et al. conceived to connect the SLTH to the Random Subset Problem.
>
> At the current stage of the SLTH theory, however, there are no rigorous theoretical results yet that show a connection between such property and ML training.
>
> To give more context, the SLTH has been motivated by empirical work such as Zhou et al. and Ramanujan et al., who showed that, leveraging gradient descent, it was possible to prune the initial random network (without changing the weights) in a way that achieves good performance. The theory of SLTH is then concerned with proving, mathematically, how powerful such training-by-pruning can be.
>
> More details on these previous works are provided in our introduction and related work sections. Interestingly, no previous work on the SLTH remarks what the reviewer is pointing out, i.e. that it would be interesting to relate the independence of the RSS gadgets leveraged in the SLTH construction to the actual algorithms by Zhou et al. and Ramanujan et al. that perform the pruning.
>
> In practice, one can expect these algorithms to achieve better practical performance by not constraining the weights in the network to be independent, but the SLTH theory, and our result in particular, suggests that the improvement can only be of logarithmic order.
>
> We conclude by emphasizing that the SLTH problem is to imitate a given trained network by pruning a larger network, according to **any** pruning strategy (including those not leveraging gradient descent). Our construction, like all previous SLTH results, does not provide a direct relationship to training via gradient descent. We believe that the latter is an interesting avenue of future research.
>
> We welcome any further request for clarification.

---

### Official Review · Reviewer_vn97 · 2025-11-01

**Soundness:** 2
**Presentation:** 2
**Contribution:** 2
**Rating:** 4
**Confidence:** 2

**Summary:**

This paper extends the theoretical foundation of the Random Subset Sum Problem (RSSP) to the quantized (discrete) domain, establishing a theoretical guarantee for the Strong Lottery Ticket Hypothesis (SLTH) under quantization. Specifically, the authors present a unified framework that provides theoretical guarantees for SLTH, encompassing both the approximate representation in the continuous setting and the exact representation in the discrete setting.

**Strengths:**

+ Table 1 plays a crucial role in highlighting the authors’ contributions relative to prior works. Specifically, it demonstrates that the proposed approach unifies both the approximate solution in the continuous setting and the exact solution in the discrete (quantized) setting with the highlight of previous study limitations. Moreover, Table 1 illustrates that the proposed method exhibits an inverse exponential decay in failure probability within the quantized setting.
+ This paper provides the stronger theoretical bound with the exponential decay in failure probability for the SLTH in the quantized setting.

**Weaknesses:**

- It would be valuable to investigate how the SLTH failure probability decays across different architectures such as ViTs, VLMs, and LLMs. Visualizing this behavior in real-world networks—with respect to factors like layer depth and width—would provide useful insights into how the likelihood of SLTH failure changes across different architectures.
- The authors have theoretically demonstrated that SLTH failure decays exponentially in the quantized setting. I strongly encourage them to conduct empirical analyses across different architectures (ViT, LLM, and VLM) to validate this behavior in practice. Such experiments would help confirm whether SLTH can indeed be identified in quantized VLMs without significant accuracy degradation and quantify the extent of model compression achieved in terms of parameter reduction.
- Although the paper establishes a theoretical bound for SLTH in the quantized setting, it lacks contributions toward identifying improved sparse models that offer practical benefits.

**Questions:**

Please refer to Weaknesses section

---

> ### Author Response · Authors · 2025-11-15
>
> We thank the reviewer for their precious feedback.
>
> SLTH has already been established for CNNs (Cunha et al., 2023) and Transformers (Otsuka et al., 2025). Our primary goal in this work was to unify the quantized and continuous settings. While the proposed approach can naturally be extended to other architectures, we leave this direction for future work.
>
> We acknowledge that the paper focuses exclusively on theoretical analysis. We respectfully refer the reviewer to our official comment, where we explain our rationale for not including empirical experiments in the original submission. We have now added an experimental component to the paper; please see the official comment for details.
>
> Please let us know if our response satisfactorily addresses all your concerns regarding our paper. If not, we would be happy to provide further clarifications.

---

### Author Response · Authors · 2025-11-15
**Response to Reviewer Concerns Regarding Experimental Validation**

Several reviewers raised concerns about the lack of experimental validation in our paper. As stated in the manuscript, our goal is to provide stronger *theoretical* guarantees that support the **existing** empirical observations.

We considered performing experiments similar to those of Cunha et al., 2023, which demonstrate that our new RSS guarantees allow a random network of a given precision to be pruned so as to closely approximate, for example, a quantized ResNet-50. However, we chose not to include such experiments in the paper because their success is essentially guaranteed: each edge of the target network corresponds to an independent RSS instance, and these instances are solved independently of one another.

We have revised the paper to include experiments in Appendix E that validate our results on discrete RSS. Theorem 2 shows that approximating a given target network by pruning a larger network is equivalent to solving a collection of independent random subset sum problems. As in Cunha et al., 2023, one may explicitly construct the desired sparse network by solving these RSS instances. Since our theoretical results specify the conditions under which RSS can be solved with high probability, and these conditions can be ensured by taking the large network to be sufficiently overparameterized, the existence of any given target network as a subnetwork is guaranteed.

We also estimate the running time for this procedure. Based on empirical measurements, solving a single RSS instance with $n = 50$ and $M = 128$ takes, on average, $7 \times 10^{-5}$ seconds (averaged over $10^5$ trials). Thus, for example, if the target network is ResNet-50, which has $23$ million parameters, the time required to find it by pruning a sufficiently large network is roughly $27$ minutes. See Appendix E for more details.

---

### Author Response · Authors · 2025-12-02
**Final Author Rebuttal**

We would like to respectfully note that much of our rebuttal focused on clarifying the established context and relevance of the Strong Lottery Ticket Hypothesis literature, rather than engaging directly with our theoretical contributions. As highlighted in the rebuttal, the theoretical concerns raised in the reviews were addressed and resolved.

The remaining points from the reviewers largely centered on requests for experiments, additional motivation, or broader contextualization. We would like to gently remark that this line of research is already well established, and our contribution is specifically aimed at **providing stronger and more unified theoretical results that build on the existing empirical findings**. We also believe that the types of experiments included in prior theoretical work are effectively guaranteed to succeed given those established results, and therefore inclusion of such kind of experiments in our paper would not provide additional insight into our theoretical contributions. For this reason, we chose not to include such experiments.

We kindly ask that this be taken into consideration when making the final decision.

---

### Meta-Review · Area_Chair_vfH3 · 2026-01-07

**Summary:**

The paper proposes a unified framework to prove the strong lottery ticket (SLT) hypothesis assuming not only discrete source but also discrete target networks.

While the problem is interesting and relevant, several reviewers criticized a lack of experimental evidence to confirm the theoretical insights. During the rebuttal, the authors added a small analysis for solving a single discrete subset sum approximation problem exactly in Appendix E, which serves to estimate the constant in the width requirement. The experiments suggest a width overparameterization factor of around 15 for M=3500. This is a bit too small to get an estimate of what source width would be required for modern architectures (ViT, LLM, and VLM), as M can also be interpreted as a quantization precision.
To see this, asking for explicit constructions of SLTs for modern architectures (ViT, LLM, and VLM) is a reasonable request by reviewers to understand whether SLTs could scale to this setting.

Furthermore, the theoretical novelty is relatively limited, as most proof strategies are quite established. The novel contribution is limited to Theorem 2, which improves the failure probability in the construction by Kumar and Natale, 2025. This is still an interesting result.

Proofs for continuous targets also capture discrete targets as special case, yet, do not handle exact reconstruction, which is a novel contribution. However, the presented results should also be obtainable from continuous target results assuming a precision of roughly 1/M. In practical settings, obtaining discrete targets also incurs a quantization error, which is considered by known results (which is also closer to how quantization in neural networks work using continuous normalization). The exact reconstruction is in a way a partial result of constructions assuming binary sources.
Yet, the construction strategy to utilize a novel subset sum approximation result is a valid contribution to the SLT literature.

**Reviewer Concerns:**

A recurring reviewer concern was a lack of experiments to support the theory.

During the rebuttal, the authors added a small analysis for solving a single discrete subset sum approximation problem exactly in Appendix E, which serves to estimate the constant in the width requirement. The experiments suggest a width overparameterization factor of around 15 for M=3500. This is a bit too small to get an estimate of what source width would be required for modern architectures (ViT, LLM, and VLM), as M can also be interpreted as a quantization precision.
To see this, asking for explicit constructions of SLTs for modern architectures (ViT, LLM, and VLM) is a reasonable request by reviewers to understand whether SLTs could scale to this setting.

There were also several issues with understanding the contribution of the paper. To cater to the general machine learning community of A* venues, it could help, if the authors could extend their discussion of the practical relevance of their results to neural network quantization. In more detail: How is M linked to the quantization precision? Why should readers care about the exact reconstructions compared to the inexact construction from binary sources in previous work?

**Reviewer Scores:**

Reviewer bZyu (current: 4) indicated that they would keep their score.

The remaining reviewers were likely to keep their scores given the lack of experiments for modern architectures.

---

### Decision · Program_Chairs · 2026-01-26

Reject